# Reservoir Computing with Spatial Filtering and Manifold Learning for fMRI Classification

## Abstract

We introduce a parametric framework that couples discriminative spatial filtering with reservoir computing to distinguish spatiotemporal structure in resting-state fMRI in two classes. Temporal dependencies are encoded in a reservoir, while supervised spatial filtering on reservoir states isolates condition-specific patterns; parametric Uniform Manifold Approximation and Projection (UMAP) then yields compact nonlinear embeddings fit on training data and evaluated with cross-subject validation. On 163 participants (97 healthy controls, 66 major depressive disorder), the method reaches 87% accuracy, outperforming network-feature pipelines using LDA, SVM, kNN, and GNN. The framework also generalizes to autism spectrum disorder classification, achieving competitive accuracy on the ABIDE (NYU) benchmark and ranking among top state-of-the-art methods. Interpretability combines spatial-pattern maps with Shapley-value attribution, providing coherent, region-level explanations that consistently implicate cortical and subcortical areas associated with both major depressive disorder and autism spectrum disorder. The framework offers an interpretable route to modeling spatiotemporal organization in clinical and cognitive fMRI.

## 1 Introduction

Classifying multivariate time series (MTS) remains a core challenge in machine learning, particularly when data is high-dimensional and discriminative features are encoded in complex spatiotemporal dynamics. Reservoir Computing (RC) has emerged as a powerful framework for modeling temporal data due to its computational efficiency and ability to capture complex system dynamics using a fixed, randomly initialized recurrent network (Zhang & Vargas, 2023; Hramov et al., 2025). However, a significant limitation of standard RC models is their handling of spatial structure within MTS. Typically, the high-dimensional sequence of reservoir states is aggregated into a single vector for classification, causing a loss of important temporal information and discarding fine-grained spatial information. Thus, it limits the performance of the model (Aswolinskiy et al., 2018; Ma et al., 2016). This is especially critical in domains like neuroimaging, where the spatial origin of signals is semantically meaningful, but the problem is fundamentally general.

Recent efforts to enhance RC have focused on better leveraging the temporal information in reservoir states. Techniques such as applying Principal Component Analysis (PCA) to the state matrix (Prater, 2017) or using the reservoir model space for unsupervised encoding (Bianchi et al., 2020) demonstrate that treating the state sequence as a structured object, rather than a flat feature vector, yields substantial gains. These approaches, however, often remain temporally-centric and do not explicitly model the spatial relationships between the channels of the input MTS. We posit that jointly modeling spatiotemporal structure, which involves coupling a temporal feature extractor with a learnable spatial transformation, is key to advancing MTS classification with RC.

In this work, we introduce a novel parametric framework that integrates Reservoir Computing with supervised spatial filtering and nonlinear manifold learning to create highly discriminative spatiotemporal representations.

We validate our framework on a challenging clinical benchmark: classifying resting-state functional magnetic resonance imaging (fMRI) data from individuals with Major Depressive Disorder (MDD)

and healthy controls. This domain exemplifies the challenges of high-dimensional spatiotemporal data where both dynamics and topography are informative. Our method achieves a state-of-the-art accuracy of 87%, significantly outperforming strong baselines including LDA, SVM, kNN, and Graph Neural Networks (GNNs) on the same dataset. The interpretability results consistently highlight brain regions known to be associated with MDD, validating the biological plausibility of the model's learned representations. To further assess generalizability, we additionally evaluated the framework on autism spectrum disorder (ASD) classification using rs-fMRI data from the ABIDE (NYU) cohort[1]. The method achieved competitive performance relative to state-of-the-art approaches, supporting its applicability beyond a single clinical condition (Xue et al., 2024)

By successfully integrating spatial filtering within the RC paradigm and leveraging parametric manifold learning, we present a general and interpretable pathway for MTS classification that effectively bridges temporal modeling with spatial optimization. The main contribution to the novelty of the developed approach is the idea of applying a supervised spatial filter (CSP) to reservoir states rather than to raw data, and an end-to-end framework that integrates the above ideas.

## 2 RELATED WORKS

### 2.1 RESERVOIR COMPUTING FOR TIME SERIES CLASSIFICATION

RC has established itself as a efficient alternative to fully-trained recurrent neural networks for temporal feature extraction due to its low training cost and strong performance on dynamical systems (Lukoševičius & Jaeger, 2009; Tanaka et al., 2019). The core idea is to project input signals into a high-dimensional space via a fixed, randomly connected reservoir, with training confined to a simple linear or nonlinear readout layer. While successful in forecasting and system identification (Andreev et al., 2022; Badarin et al., 2024), the application of standard Echo State Networks (ESNs) to time series classification often lags behind other state-of-the-art methods (Aswolinskiy et al., 2018). A key reason is the common practice of collapsing the temporal information into a single vector, typically by using the final reservoir state or averaging states over time, which discards potentially discriminative dynamical features (Aswolinskiy et al., 2018; Ma et al., 2016). To address this, several advanced representations of reservoir states have been proposed. For instance, Prater (2017) applied PCA to the reservoir state matrix to create a more informative feature set for classification. Similarly, the concept of the "reservoir model space" (Bianchi et al., 2020) provides an unsupervised encoding of the dynamics by fitting linear predictive models to reservoir state sequences. These approaches demonstrate that treating the reservoir's trajectory as a structured object is superior to simple aggregation. Our work builds on this principle but extends it by explicitly incorporating a learnable spatial transformation optimized for classification.

### 2.2 SPATIAL FILTERING AND SPATIOTEMPORAL MODELING

Spatial filtering techniques are designed to enhance the signal-to-noise ratio and discriminative power of multichannel data by projecting it into a space where class differences are maximized. The Common Spatial Patterns (CSP) algorithm is a cornerstone of this approach, widely used in EEG-based brain-computer interfaces to find spatial filters that maximize the variance of one class while minimizing it for another (Ramoser et al., 2000). While highly effective for signals like EEG, its application has been largely separate from deep temporal modeling. The integration of spatial filtering with recurrent networks has been explored in limited contexts, such as for EEG classification (Lawhern et al., 2018), but these approaches typically involve training the entire network end-to-end, forfeiting the computational efficiency of RC. A few studies have combined RC and CSP, but often in a pipeline where CSP is applied as a static pre-processing step to the raw signals before temporal modeling.

### 2.3 MANIFOLD LEARNING AND DIMENSIONALITY REDUCTION

Dimensionality reduction is a critical step for handling the high-dimensional features generated by RC and spatial filtering. While PCA is a common linear approach (Prater, 2017), nonlinear techniques can capture more complex, hierarchical structures. Manifold learning methods like t-SNE

---

[1]https://fcon_1000.projects.nitrc.org/indi/abide/

and UMAP (Uniform Manifold Approximation and Projection) have gained prominence for their ability to preserve both local and global data topology in a low-dimensional embedding (Maaten & Hinton, 2008; McInnes et al., 2018). UMAP, in particular, is grounded in Riemannian geometry and algebraic topology (McInnes et al., 2018). It approximates the local Riemannian structure of data by constructing a weighted $k$-nearest neighbor graph formalized as a fuzzy simplicial set, and defines a corresponding fuzzy representation in the embedding space. The low-dimensional embedding is then optimized by minimizing the cross-entropy between the two representations using stochastic gradient descent with negative sampling. This procedure preserves both local neighborhoods and aspects of the global structure, yielding embeddings that are topologically faithful and computationally efficient. Compared to t-SNE, UMAP has been shown to better retain global organization while providing competitive local clustering performance.

However, a significant limitation of standard non-parametric UMAP is that it does not produce a transform function that can be applied to new, unseen data, making it unsuitable for a standard train-test validation pipeline. Recent developments have introduced parametric UMAP, which uses a neural network to learn the mapping from the high-dimensional space to the embedding manifold (Sainburg et al., 2021). This allows for out-of-sample extension, a feature we leverage for the first time in an RC-based classification framework. While UMAP is widely used for visualization, its application as a parametric, trainable component within a classification model for spatiotemporal data is novel. Our use of parametric UMAP to project test data into a manifold learned from training data provides a powerful and generalizable nonlinear alternative to linear readout layers like LDA or SVM.

## 2.4 NEUROIMAGING AND CLINICAL CLASSIFICATION

In the specific application domain of neuroimaging, MDD and ASD classification has been attempted using various methods. Traditional approaches often rely on features derived from functional connectivity networks, which are then classified using standard machine learning models like SVM (Andreev et al., 2023) or more recently, GNNs (Pitsik et al., 2023). While these network-based methods are effective, they fundamentally analyze static summaries of brain activity, potentially overlooking the temporal dynamics that are central to brain function. Deep learning models, including Convolutional Neural Networks (CNNs) applied to connectivity matrices (Pilmeyer et al., 2022) and recurrent models applied to time series (Zhang et al., 2023), have also been explored. For ASD classification the researchers propose analyze functional connectivity networks by using recurrent neural networks (RNN)-based models, Fourier transform-based methods, GNNs and transformers (Xue et al., 2024; Wang et al., 2019; Bedel et al., 2023; Li et al., 2021). However, these models often require large amounts of data and are computationally intensive to train. Our RC-based approach offers a computationally efficient alternative that directly models the raw spatiotemporal signals. The interpretability of our model, achieved through spatial patterns and Shapley values, also addresses a key shortcoming of many deep learning models in providing clinically plausible explanations.

In summary, our work distinguishes itself by unifying RC, supervised spatial filtering on reservoir states, and parametric UMAP into a single, end-to-end framework. This approach effectively bridges the gap between sophisticated temporal modeling and optimized spatial feature extraction, providing a novel and interpretable solution for MTS classification that is validated on a clinically relevant problem.

## 3 METHOD OVERVIEW

We propose a new multistage approach schematically illustrated in Fig. 1 combining different machine learning methods as CSP for capturing spatial features of the data, RC for temporal features, UMAP for nonlinear dimensionality reduction, and Linear Discriminant Analysis (LDA) for classification. The main contribution to the novelty of the developed approach is the idea of applying a supervised spatial filter (CSP) to reservoir states rather than to raw data, and an end-to-end framework that integrates the above ideas.

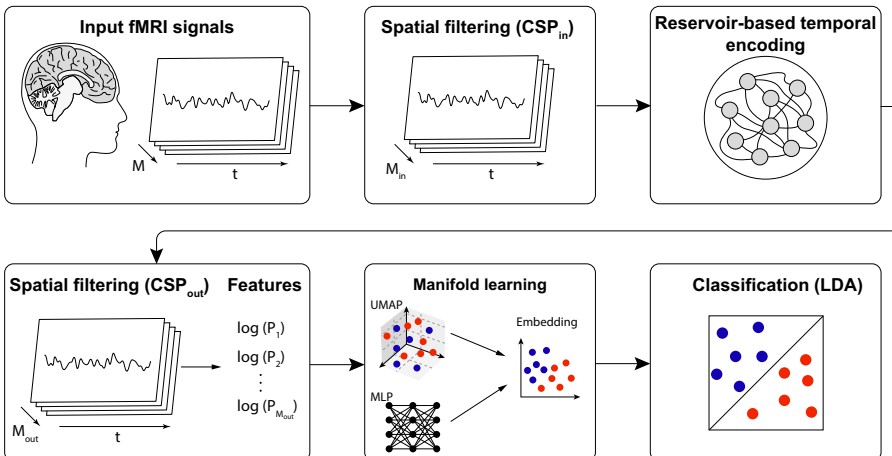

Figure 1: **Schematic representation of the proposed approaches for classification of multivariate BOLD signals.** As an input data we use $M$ BOLD signals associated with brain regions. A combined approach involving preliminary feature extraction using CSP, their nonlinear transformation using RC to account for complex temporal dependencies, reapplication of CSP to refine the spatial features, manifold learning for dimensionality reduction, and final classification via LDA.

### 3.1 STAGE I (CSP-I)

As the first step, CSP is applied to the raw BOLD of shape $K \times T$ signals to extract the $M_{inp}$ most discriminative spatial components. This step acts as a smart dimensionality reduction, filtering out noisy or non-informative spatial dimensions and presenting the reservoir with a purified, feature-rich input.

CSP algorithm is a supervised spatial filtering technique used to maximize the variance of signals from one class while minimizing the variance for another (Chacon-Murguia et al., 2020).

For a set of trials $\mathbf{X}_j^c$ (where $c$ denotes the class, $K$ is the number of channels, and $T$ is time), the normalized spatial covariance matrix for each class is computed:

$$\mathbf{R}^c = \frac{1}{P_c} \sum_{j=1}^{P_c} \frac{\mathbf{X}_j^c (\mathbf{X}_j^c)^T}{\text{trace}(\mathbf{X}_j^c (\mathbf{X}_j^c)^T)}. \tag{1}$$

The composite covariance $\mathbf{R} = \mathbf{R}^1 + \mathbf{R}^2$ is factorized ($\mathbf{R} = \mathbf{U}\mathbf{\Lambda}\mathbf{U}^T$) and whitened ($\mathbf{P} = \mathbf{\Lambda}^{-1/2}\mathbf{U}^T$). The whitened matrices $\mathbf{S}^1 = \mathbf{P}\mathbf{R}^1\mathbf{P}^T$ and $\mathbf{S}^2 = \mathbf{P}\mathbf{R}^2\mathbf{P}^T$ share common eigenvectors $\mathbf{B}$, such that $\mathbf{S}^1 = \mathbf{B}\mathbf{\Lambda}_1\mathbf{B}^T$ and $\mathbf{\Lambda}_1 + \mathbf{\Lambda}_2 = \mathbf{I}$.

The CSP projection matrix is $\mathbf{F} = \mathbf{B}^T\mathbf{P}$. For a trial $\mathbf{X}$, the spatially filtered signal is $\mathbf{Z} = \mathbf{F}_{M_1}^T\mathbf{X}$, where $\mathbf{F}_{M_1}$ contains the first and last $M_1$ filters. The final feature vector for classification is the log-variance of these components:

$$p_m = \log\left(\frac{1}{T}\sum_{t=1}^{T} z_{m,t}^2\right), \quad \text{for } m = 1, ..., M_1. \tag{2}$$

The optimal number of filters $M_{inp} = 14$ was determined via cross-validation. A small regularization parameter ($\alpha = 10^{-3}$) was added to the diagonal of $\mathbf{R}^c$ to ensure numerical stability.

### 3.2 STAGE II (RC)

Then, the obtained $M_1$ signals are submitted to an RC. The RC architecture is a versatile platform for processing time series and solving classification and prediction problems. It includes three key components: an input layer that accepts input data from external sources; a reservoir layer consisting of a large number of recurrent neurons randomly connected to each other, which provides the

transformation of input data into a high-dimensional state space; and an output layer that performs reservoir state analysis and generates final predictions or classifications.

We propose a reservoir configuration that is characterized by a complete spatial separation of the reservoir inputs. This architecture allows us to take into account the specificity of the data and significantly improve the classification accuracy, which makes this approach promising for analyzing neurophysiological data. The approach is inherited from Hramov et al. (2024), where it was proposed to use a complete separation of reservoir inputs between different artificial neurons within a reservoir hidden inner layer to predict the dynamics of a stochastic system. In other words, each neuron in the reservoir layer was connected with only one input.

The RC transforms input time series via a fixed, randomly initialized recurrent network (the "reservoir") of size $N_h$. The internal state $\mathbf{h}_t$ is updated as:

$$\mathbf{h}_t = (1 - \gamma)\mathbf{h}_{t-1} + \gamma \tanh(\mathbf{W}\mathbf{h}_{t-1} + \mathbf{G}\mathbf{g}_t + \mathbf{b})), \tag{3}$$

where $\mathbf{h}_t \in \mathbb{R}^{N_h}$ is the reservoir state at time $t$; $\mathbf{g}_t \in \mathbb{R}^K$ is the input vector; $\mathbf{W} \in \mathbb{R}^{N_h \times N_h}$ is the sparse, randomly initialized recurrent weight matrix, scaled to have a specific spectral radius $\rho$; $\mathbf{G} \in \mathbb{R}^{N_h \times K}$ is the input weight matrix, structured to enforce anatomical or component-based segregation; $\gamma \in (0, 1]$ is the leakage rate, controlling the integration of new input with the previous state; $\mathbf{b}$ is a bias term, which we take equal to 0. The hyperparameters $\rho$ and $\gamma$ are critical and were optimized for each approach.

The key innovation is the structure of $\mathbf{G}$, which ensures that inputs from specific CSP components project only onto their designated neuron subgroups, preserving spatial or feature-based specificity.

As a result of Stage II, we obtain $N_h$ signals with time length $T$.

### 3.3 STAGE III (CSP-II)

As the third step, the RC's signals are submitted to the second CSP filer ($CSP_{out}$) for Spatio-Temporal Refinement of features after nonlinear transformation by reservoir. This step extracts the ultimate set of features that are maximally discriminative in the combined spatial-temporal feature space created by the reservoir. Unlike the Step II, we obtain $M_{out} = 20$ features.

### 3.4 STAGE IIIo (UMAP)

Additionally, after $CSP_{out}$, we apply Parametric UMAP (P-UMAP) for nonlinear dimensionality reduction. P-UMAP extends UMAP by training a neural network with the same objective, which enables consistent out-of-sample projection of test data. This step reduces the high-dimensional $CSP_{out}$ features to a compact embedding, ensuring stable representation and improved class separability for the final LDA classifier (see Fig. 1). In practice, the $M_{out}$ features are compressed to 2–5 components.

### 3.5 STAGE IV (LDA)

On the final stage we apply LDA for classification of the log-variance of the obtained features. As a solver we use singular value decomposition (svd).

### 3.6 INTERPRETABILITY

To move beyond a black-box model, we developed a method to identify which original brain regions contributed most to the classification decision in the proposed approach.

We computed SHAP (SHapley Additive exPlanations) values for the classifier combining P-UMAP and LDA. This approach quantified the contribution of each $M_{out}$ CSP features within the low-dimensional embedding to the final classification outcome. The absolute values of the spatial pattern matrix (the inverse of the CSP filter matrix) were used to project the SHAP values backwards onto the reservoir neurons, estimating each neuron's importance. The importance of the reservoir neurons was then averaged according to the input segregation defined by matrix $\mathbf{G}$, yielding the importance of each of the initial $M_{inp}$ CSP components. Finally, the absolute values of the spatial pattern matrix

from the first CSP step were used to project these component importances back to the original 139 AAL3 brain regions, resulting in a final significance vector $\mathbf{S}^{SL}$ for each ROI.

The detailed description of the interpretability approach is provided in the Appendix A.

This backward propagation provides a transparent, quantitative link between the model's performance and the underlying neuroanatomy.

## 4 BENCHMARKS

To contextualize our model's performance, we implemented a series of established benchmark methods based on functional connectivity for MDD dataset. The benchmark methods are chosen according to the literature as the most common ones for classification of fMRI data (Bondi et al., 2023; Pilmeyer et al., 2022; Pisarchik et al., 2023; Pitsik et al., 2023; Andreev et al., 2023):

1. **Functional Connectivity (FC) Matrices:** For each subject, a 139×139 symmetric FC matrix was constructed by calculating the Pearson correlation coefficient between every pair of BOLD time series.
2. **Graph Theory Measures:** Five global network metrics were extracted from each thresholded FC matrix: Mean Node Strength, Average Shortest Path Length, Clustering Coefficient, Small-World Coefficient, and Number of Edges.
3. **Classical ML Classifiers:** These five graph measures served as input features for an LDA, a Support Vector Machine (SVM) with RBF kernel, and a k-Nearest Neighbors (kNN) classifier. Hyperparameters were optimized via grid search.
4. **Graph Neural Network (GNN):** A two-layer graph convolutional network was implemented to operate directly on the FC matrix, representing a modern deep learning approach for graph-structured data.

All models were evaluated using a robust 100-iteration stratified shuffle-split cross-validation procedure. The detailed description of the benchmark methods is provided in the Appendix C.

## 5 EXPERIMENTS

### 5.1 MAJOR DEPRESSIVE DISORDER CLASSIFICATION

#### 5.1.1 MDD DATASET

We analyze a dataset of $P = 163$ participants, comprising $P_{HC} = 97$ healthy controls (HC) and $P_{MDD} = 66$ patients diagnosed with major depressive disorder (MDD). All MDD diagnoses were confirmed by experienced psychiatrists using the Mini International Neuropsychiatric Interview (MINI) (Sheehan et al., 1998), with symptom severity assessed by the Montgomery–Åsberg Depression Rating Scale (MADRS) (Montgomery & Åsberg, 1979; Müller et al., 2003). Exclusion criteria for all participants included a history of comorbid psychiatric conditions, neurological disorders, significant head trauma, or standard MRI contraindications. The two groups were matched for age, gender, and education level, while, as expected, the MDD group exhibited significantly higher MADRS scores (see Table 1). The study was approved by the Ethical Committee of the Medical University of Plovdiv (Approval No: 2/19.04.2018), and all participants provided written informed consent.

All MRI data were acquired on a 3T GE Discovery 750w scanner. For each participant, we obtained a high-resolution T1-weighted anatomical scan and a resting-state fMRI (rs-fMRI) scan. The rs-fMRI parameters were: repetition time TR = 2000 ms, echo time TE = 30 ms, flip angle = $90°$, 3 mm slice thickness, matrix size of $64 \times 64$, and 192 volumes (Stoyanov et al., 2020).

Preprocessing was performed using SPM12 and included standard steps: motion correction, co-registration of functional images to the individual's T1-weighted scan, and spatial normalization to the Montreal Neurological Institute (MNI) template (Pitsik et al., 2023; Stoyanov et al., 2022).

The preprocessed brain volume was parcellated into $K = 139$ regions using the Automated Anatomical Labeling (AAL3) atlas (Rolls et al., 2020). This yielded a multivariate time series for each par-

Table 1: The demographic and clinical parameters of the participant samples were assessed. The two cohorts included a healthy control group (HC) and a group of individuals with major depressive disorder (MDD).

|  | HC ($n = 94$) | MDD ($n = 70$) | Significance |
|---|---|---|---|
| Age (mean $\pm$ SD) | $40.6 \pm 11.8$ | $41.0 \pm 13.2$ | $0.961^a$ |
| Education (secondary/higher) | 5/89 | 7/63 | $> 0.999^b$ |
| Sex (M/F) | 41/53 | 26/44 | $0.996^b$ |
| MADRS score (mean $\pm$ SD) | $2.0 \pm 2.6$ | $29.5 \pm 6.0$ | $< 0.001^a$ |

MADRS — Montgomery–Åsberg Depression Rating Scale, SD — Standard Deviation, [a]Two-sample Kolmogorov-Smirnov nonparametric test, [b]$\chi^2$ — test.

ticipant, represented as a matrix $\mathbf{X}_j^c \in \mathbb{R}^{K \times T}$, where $c \in \text{HC}, \text{MDD}$ denotes the class label, $j$ is the participant index, $K = 139$ is the number of regions (channels), and $T = 192$ is the number of time points.

### 5.1.2 COMPARISON

We employed Stratified Shuffle Split cross-validation to divide subjects into 10 randomized folds with a fixed test subject proportion of 20% ensuring that all time series from a single subject were assigned to either the training or test set on each split (subject-wise cross-validation). Stratification was used to ensure proportional representation of the MDD class in both training and testing subsets relative to the overall dataset. Consequently, each training fold comprised 130 subjects, with the MDD subgroup constituting 40.77%. Correspondingly, each independent test fold contained 33 subjects, with the MDD prevalence held at 39.40%.

Model performance was quantified using classification accuracy on each of the 10 test folds. The resulting 10 accuracy scores were subsequently averaged to compute the mean accuracy which was used as a final evaluation metric.

We analyzed the effect of reservoir parameters (leak rate and spectral radius which are optimized by grid search in ranges from 0.1 to 0.5 with step 0.05 and from 0.8 to 1.1 with step 0.05 respectively) on classification accuracy as well as the contribution of applying UMAP (Fig. 2). Each point in the figure corresponds to a particular reservoir configuration, with accuracy values averaged over 10 cross-validation folds. The results demonstrate that UMAP consistently improves both stability and overall accuracy compared to the pipeline without dimensionality reduction (paired samples t-test:

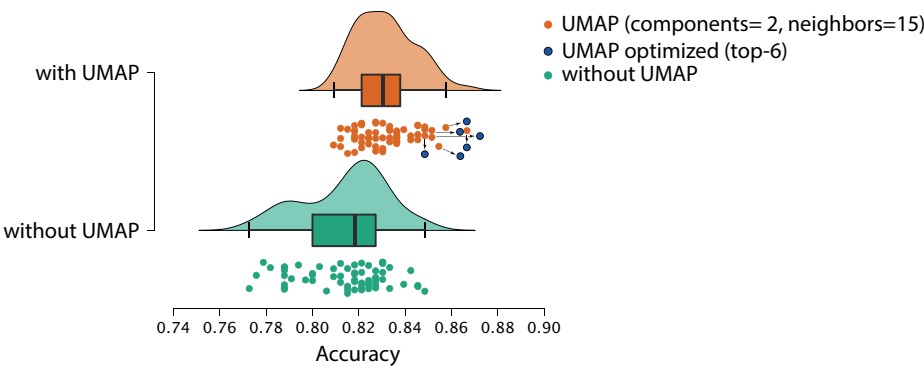

Figure 2: **Effect of UMAP on classification accuracy across reservoir configurations.** Distribution of accuracy values obtained for different reservoir parameters (leak rate, spectral radius) with and without UMAP. Orange: P-UMAP with fixed parameters (2 components, 15 neighbors); blue: top-6 optimized P-UMAP configurations; green: pipeline without UMAP.

$t = 5.78$, $df = 62$, $p = 2.6 \times 10^{-7}$). For the six best-performing reservoir configurations (top 10% of all configurations), we further optimized UMAP hyperparameters: the number of neighbors $N_n \in \{5, 10, 15, 20\}$ and the number of output components $N_c \in \{2, 3, 4, 5\}$. These results are shown in Table 2 and in Fig. 2 as blue points, with arrows indicating changes in accuracy after UMAP optimization for each configuration. As one can see, optimizing UMAP parameters yields an additional gain in accuracy and further stabilizes the results.

We also investigated the influence of key hyperparameters on model performance, including number of input and output CSP components ($M_{inp}$, $M_{out}$), leak rate ($\gamma$), spectral radius ($\rho$). The detailed analysis is described in Appendix B. Our analysis revealed that initial increases in both $M_{inp}$ and $M_{out}$ progressively enhanced accuracy, with optimal performance achieved at $M_{inp} = 14$ and $M_{out} = 20$. Beyond these values, further increments did not improve pipeline performance, indicating saturation of beneficial effects. For the reservoir dynamics, we identified that leak rate and spectral radius do not greatly affect the classification accuracy: mean accuracy varies in range [0.82, 0.85] for whole investigated ranges of parameters.

We also compare our approach with the standard benchmark methods (Sec. 4). The classification results are summarized in Table 3. Our full approach with fixed UMAP parameters $N_n = 15$, $N_c = 2$ achieved a mean accuracy of 87%, significantly outperforming most of other methods. One can see, approach without UMAP (CSP-RC-CSP-LDA) shows similar accuracy of 85%, but recall is significantly lower. Simple approach without RC demonstrates the lowest accuracy across all the considered methods. Among benchmark methods, GNN shows the best results but significantly lower that our approach.

The fact that our model substantially exceeded the performance of all benchmark methods, including the GNN, underscores that directly modeling the raw spatiotemporal signal can be more powerful than relying on pre-computed functional connectivity summaries.

### 5.1.3 INTERPRETABILITY

We have identified a number of regions with significant difference between MDD and HC groups (Fig. 4). One of them, the medial superior frontal gyrus, serves as a central hub of the default mode network (DMN), which has been repeatedly linked to heightened self-referential thinking and maladaptive rumination in individuals with major depressive disorder (MDD) (Sheline et al., 2010; Whitfield-Gabrieli & Ford, 2012; Greicius et al., 2007). Heightened activity in this region may promote an excessive preoccupation with negative internal thoughts—a core feature of depression.

Changes were also noted in areas involved in primary and early visual processing, such as the calcarine sulcus, inferior occipital gyrus, and the lateral geniculate nucleus (LGN). These structures contribute to basic visual perception and the filtering of attentional signals. The LGN, which acts as a thalamic relay for visual input, interfaces with cortical attentional and salience networks to regulate the transmission of emotionally salient visual information (Portas et al., 1998; Pessoa & Adolphs, 2010).

Another region of significance is the right paracentral lobule, which exhibited increased cortical thickness in patients with major depressive disorder (MDD). This area has been implicated in the

Table 2: Results of UMAP optimization.

| Metric | top-1 | top-2 | top-3 | top-4 | top-5 | top-6 |
|---|---|---|---|---|---|---|
| Accuracy before optimization | **0.87** | 0.86 | 0.85 | 0.85 | 0.85 | **0.85** |
| Accuracy after optimization | **0.87** | **0.87** | **0.87** | **0.87** | 0.86 | **0.85** |
| **Hyperparameters after optimization** | | | | | | |
| number of neighbors $N_n$ | 15 | 5 | 5 | 5 | 20 | 10 |
| number of output components $N_c$ | 2 | 4 | 4 | 2 | 4 | 2 |
| leak rate $\gamma$ | 0.1 | 0.2 | 0.1 | 0.2 | 0.25 | 0.45 |
| spectral radius $\rho$ | 0.95 | 0.8 | 0.9 | 0.85 | 1.1 | 0.95 |
| number of filters in the first CSP $M_{inp}$ | 14 | 14 | 14 | 14 | 14 | 14 |
| number of filters in the second CSP $M_{out}$ | 20 | 20 | 20 | 20 | 20 | 20 |

Table 3: Classification performance metrics for MDD dataset.

| Method | Accuracy | Recall | Precision | F1-Score |
|---|---|---|---|---|
| **CSP-RC-CSP-UMAP-LDA** | $0.87 \pm 0.05$ | $0.80 \pm 0.10$ | $0.88 \pm 0.09$ | $0.83 \pm 0.07$ |
| CSP-RC-CSP-KPCA-LDA | $0.86 \pm 0.05$ | $0.72 \pm 0.12$ | $0.92 \pm 0.09$ | $0.80 \pm 0.08$ |
| CSP-RC-CSP-PCA-LDA | $0.86 \pm 0.05$ | $0.72 \pm 0.10$ | $0.92 \pm 0.08$ | $0.80 \pm 0.07$ |
| CSP-RC-UMAP-LDA | $0.86 \pm 0.06$ | $0.78 \pm 0.11$ | $0.85 \pm 0.08$ | $0.81 \pm 0.08$ |
| CSP-RC-CSP-LDA | $0.85 \pm 0.05$ | $0.72 \pm 0.10$ | $0.87 \pm 0.07$ | $0.79 \pm 0.08$ |
| RC-UMAP-LDA | $0.59 \pm 0.06$ | $0.2 \pm 0.15$ | $0.44 \pm 0.17$ | $0.25 \pm 0.16$ |
| RC-CSP-UMAP-LDA | $0.57 \pm 0.10$ | $0.16 \pm 0.12$ | $0.44 \pm 0.33$ | $0.22 \pm 0.16$ |
| CSP-UMAP-LDA | $0.55 \pm 0.06$ | $0.38 \pm 0.09$ | $0.42 \pm 0.09$ | $0.39 \pm 0.07$ |
| **Benchmarks** | | | | |
| GNN (on FC matrix) | $0.64 \pm 0.11$ | $0.55 \pm 0.15$ | $0.59 \pm 0.14$ | $0.56 \pm 0.12$ |
| LDA (on graph metrics) | $0.60 \pm 0.06$ | $0.32 \pm 0.09$ | $0.54 \pm 0.09$ | $0.39 \pm 0.08$ |
| kNN (on graph metrics) | $0.56 \pm 0.06$ | $0.45 \pm 0.09$ | $0.47 \pm 0.08$ | $0.46 \pm 0.08$ |
| SVM (on graph metrics) | $0.58 \pm 0.03$ | $0.08 \pm 0.04$ | $0.45 \pm 0.20$ | $0.14 \pm 0.07$ |

interpretation of sensory information, including the perception of emotions in facial stimuli (Peng et al., 2015).

## 5.2 AUTISM SPECTRUM DISORDER CLASSIFICATION

### 5.2.1 ASD DATASET

We have tested our approach additionally on the fMRI data from the NYU site of public ABIDE dataset[2] which is widely used as a benchmark fMRI dataset. Original ABIDE dataset comprises of $P = 289$ participants, with $P_{HC} = 135$ healthy controls and $P_{ASD} = 154$ subjects with autism spectrum disorder (ASD). For our study, we discarded data of three patients with ASD due to insufficient data quality in most of their fMRI channels, leaving us with $P = 286$ total subjects and $P_{ASD} = 151$ ASD patients. All rs-fMRI data were acquired based on a standard echo-planar imaging sequence on a clinical routine 3.0T Allegra scanner with the following imaging parameters: TR/TE is 2000 ms/15 ms with 180 volumes, the number of slices is 33, and the slice thickness is 4.0 mm.

### 5.2.2 COMPARISON

As for MDD dataset, we employed Stratified Shuffle Split cross-validation to divide subjects into 10 randomized folds with a fixed test subject proportion of 20% ensuring that all time series from a single subject were assigned to either the training or test set on each split (subject-wise cross-validation). Stratification was used to ensure proportional representation of the MDD class in both training and testing subsets relative to the overall dataset. Consequently, each training fold comprised 228 subjects, with the ASD subgroup constituting 52.63%. Correspondingly, each independent test fold contained 58 subjects, with the ASD prevalence held at 53.45%.

Model performance was quantified using classification accuracy on each of the 10 test folds. The resulting 10 accuracy scores were subsequently averaged to compute the mean accuracy which was used as a final evaluation metric.

We optimized our approach by using grid search for RC parameters (leak rate and spectral radius) while all others were fixed. We choose the configuration which provides the best accuracy across all random folds on the test parts. The obtained results are shown in Table 4.

As the benchmark results for the NYU site of ABIDE dataset we use the materials from the paper of Xue et al. (2024). As one can see from Table 4, our approach outperforms the most of the SOTA methods: only 3 methods (dCSL, BolT, and CRNN) demonstrate higher accuracy. Moreover, F1-score of our approach is higher than almost all of them (only DART achieved 73%). The recall and precision values of our approach are also at top-3.

---

[2]https://fcon_1000.projects.nitrc.org/indi/abide/

Table 4: Classification performance metrics for ASD dataset. Best results are highlighted with ***, suboptimal results are highlighted with **, and subsuboptimal results are highlighted with *.

| Method | Accuracy | Recall | Precision | F1-Score |
|---|---|---|---|---|
| **CSP-RC-CSP-UMAP-LDA** | 0.67 | 0.71* | 0.69* | 0.70** |
| **Benchmarks**(Xue et al., 2024) | | | | |
| DART | 0.65 | 0.65 | / | 0.73*** |
| dCSL | 0.72*** | 0.68 | 0.70** | 0.69* |
| BolT | 0.71** | 0.64 | 0.72*** | 0.66 |
| BrainGNN | 0.56 | 0.79*** | 0.44 | 0.57 |
| CRNN | 0.68* | 0.66 | 0.66 | 0.65 |
| BrainTGL | 0.58 | 0.72** | 0.33 | 0.43 |
| BrainGB | 0.58 | 0.54 | 0.58 | 0.55 |
| MVS-GCN | 0.50 | 0.36 | 0.47 | 0.49 |
| MDGL | 0.61 | 0.60 | 0.58 | 0.58 |
| GRU | 0.61 | 0.61 | 0.58 | 0.59 |
| LSTM | 0.63 | 0.61 | 0.59 | 0.59 |
| RFF | 0.66 | 0.65 | 0.62 | 0.63 |
| CRNN | 0.68 | 0.66 | 0.66 | 0.65 |
| SA-CRN | 0.62 | 0.61 | 0.60 | 0.60 |
| MDGL | 0.61 | 0.60 | 0.58 | 0.58 |
| SHeC | 0.57 | 0.55 | 0.55 | 0.55 |

### 5.2.3 INTERPRETABILITY

We have identified 9 fMRI regions with significant differences between ASD and HC groups (see Fig. 5). Notably, 5 of these regions are located in the cerebellum, one of the most consistent sites of abnormality in ASD patients (D'Mello & Stoodley, 2015). Four of them constitute right and left cerebellar Crus I/II, a region shown to have reduced gray matter volume and decreased fractional anisotropy in white matter tracks connecting this region to the dentate nucleus and contralateral cerebral cortex, structural deficits correlated with impaired communication, social interaction deficits, and repetitive behaviors (D'Mello & Stoodley, 2015). The fifth significant cerebellum region is lobule X, shown to have hyper-connectivity in ASD patients, which may contribute to atypical visual exploration and eye movement control difficulties (Lanciano et al., 2025).

Three other significant regions belong to the orbitofrontal cortex. Laminar-specific imbalances in excitatory and inhibitory circuits in the orbitofrontal cortex are observed in ASD patients. Myelinated axons, proxies for excitatory pathways, show reduced density and diameter across layers in ASD, alongside lower excitatory neuron density. These changes likely disrupt OFC communications with limbic cortices and the amygdala, providing an anatomic basis for social interaction and emotional deficits in ASD (Liu et al., 2020).

The last detected channel with significant difference between HC and ASD is the right middle temporal pole, a region where increased cortical thickness was associated with more severe communication impairment in ASD patients (Pereira et al., 2018).

## 6 CONCLUSIONS

We proposed a multistage framework that integrates supervised spatial filtering, reservoir computing, and parametric UMAP for fMRI classification. The approach effectively captures both temporal dependencies and spatial structure, achieving state-of-the-art accuracy in distinguishing MDD and ASD patients from healthy controls. Our analysis shows that UMAP improves stability and accuracy across reservoir configurations, while interpretability analysis provides biologically plausible insights into disorder-related brain regions. The results highlight the potential of combining reservoir computing with manifold learning as a general strategy for modeling high-dimensional spatiotemporal signals.

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

# A  INTERPRETABILITY

A methodological framework was developed to assess the contribution of each input fMRI channel to the classification outcome, considering all stages of data transformation [see Fig. 3]. The significance evaluation is conducted sequentially, beginning with the final stage of the model and progressing toward the original input signals. This sequential approach ensures a systematic examination of feature influence at each transformation stage, thereby preserving model interpretability. Consequently, the framework facilitates the identification of the most discriminative fMRI channels for classification, which in turn highlights the brain regions whose BOLD signals are most critical for patient classification.

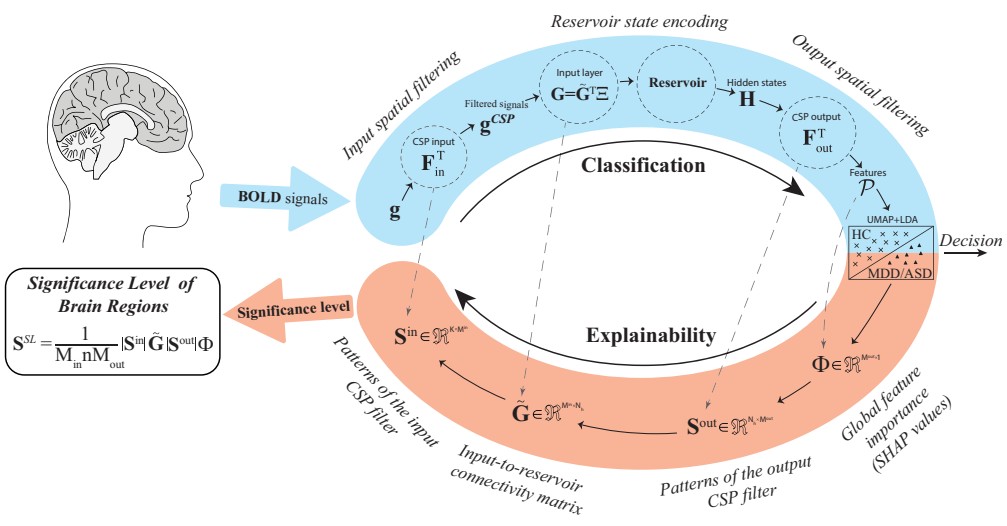

Figure 3: **Framework for assessing the contribution of fMRI channels to MDD or ASD classification.** Schematic representation of the sequential procedure to evaluate feature significance, starting from the final LDA classifier and propagating backward through the model's transformation stages (UMAP, reservoir states, CSP filters) to the original multivariate fMRI input signals. This methodology ensures interpretability by identifying discriminative brain regions critical for distinguishing MDD or ASD patients from healthy controls.

The contribution of input BOLD signals to the classification outcome is quantified at the final stage of the model, which consists of the parametric UMAP–LDA pipeline, by analysing how features extracted from the reservoir influence the learned embedding and its resulting decisions. To assess this contribution, the SHAP (SHapley Additive exPlanations) method was employed (Lundberg & Lee, 2017; Lundberg et al., 2018). It should be noted, however, that applying the SHAP method directly to raw multivariate time-series inputs presents challenges, as individual time points are not independent features but rather components of a temporally correlated sequence. Furthermore, estimating the contribution of an entire time series using SHAP introduces complexities in appropriately masking temporal dependencies. Despite the generalizability of the SHAP framework, these considerations necessitate the development of a specialized procedure for channel contribution estimation that accounts for the specific characteristics of the model. Accordingly, SHAP values ($\mathbf{\Phi}$) were computed for the final parametric UMAP–LDA stage of the model, using the training dataset to quantify how features emerging from the spatially filtered reservoir states ($\mathbf{F}_{out}^{T}$) contribute to the learned UMAP embedding and, consequently, to the classification decision.

The importance of each reservoir neuron is quantified by computing a weighted average of the absolute values of the spatial filter patterns $\mathbf{S}^{out}$, where the weights correspond to the SHAP values. This yields a vector $\bar{\mathbf{S}}^{out}$ which represents the relative contribution of each neuron to the final classification outcome. Formally, this is expressed as:

$$\bar{\mathbf{S}}^{out} = \frac{1}{M_{out}} \left| \mathbf{S}^{out} \right| \mathbf{\Phi} \tag{4}$$

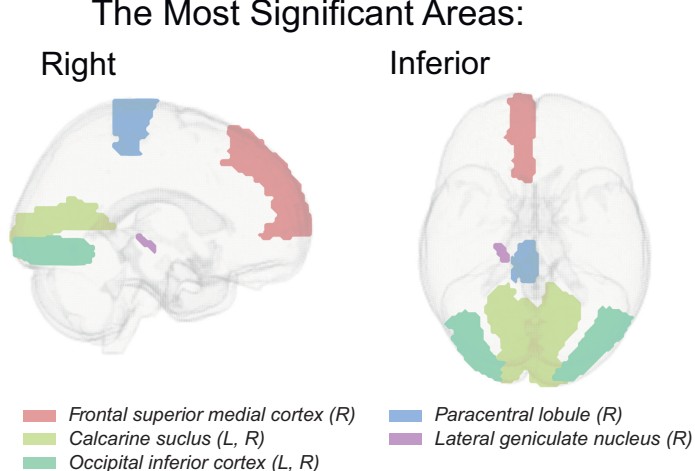

Figure 4: **Feature extraction.** Illustration of the most significant brain regions that influence the clustering process of patients with major depressive disorder.

where $\boldsymbol{\Phi}$ denotes the vector of SHAP values, quantifying the absolute contribution of each feature to the model's prediction, averaged across the training set., $M_{out}$ represents the number of output components in the spatial filter $\mathbf{F}_{out}^{T}$ used for classification.

The spatial patterns $\mathbf{S}^{out}$ corresponding to filters $\mathbf{F}^{out}$ can be interpreted as spatial maps showing the contribution of each reservoir neuron signal fed to the filter input to the formation of the corresponding component. The coefficients of the pattern $\mathbf{S}^{out}$ indicate the degree to which each reservoir neuron signal influences the resulting signal after the filter is applied, i.e., they show the extent to which the dynamics of each neuron contributes to the extraction of distinctive features necessary for class distinction.

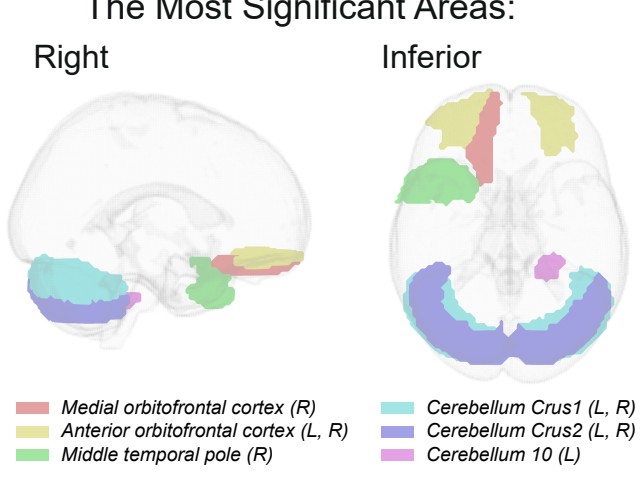

Figure 5: **Feature extraction.** Illustration of the most significant brain regions that influence the clustering process of patients with autism spectrum disorder.

The significance coefficients of the reservoir neurons are then averaged within the regions associated with specific input components $\mathbf{g}^{csp}$, allowing us to estimate their "average significance" in the transformations occurring in the $\bar{\bar{\mathbf{\Phi}}}^h$ reservoir:

$$\bar{\bar{\mathbf{\Phi}}}^h = \frac{1}{n}\tilde{\mathbf{G}}\bar{\mathbf{S}}^{out}, \tag{5}$$

where $n$ is the factor of increasing the dimensionality of the input data in the high-dimensional reservoir space, and $\tilde{\mathbf{G}}$ is defined as:

$$\tilde{\mathbf{G}} = \mathbf{G}^{(RC,I_1)} \oplus \mathbf{G}^{(RC,I_2)} \oplus \cdots \oplus \mathbf{G}^{(RC,I_K)}, \tag{6}$$

where the operator $\oplus$ denotes the concatenation of matrices. Each matrix, $\mathbf{G}^{(RC,I_k)}$, defines the connections between the input signals belonging to brain area $I_k|k = \overline{1,K}$, and the $J^{(RC,I_k)}$ neurons of the hidden layer of the reservoir corresponding to that brain region.

Finally, a weighted averaging of the absolute values of the spatial patterns $\mathbf{S}^{in}$ of the input filter, corresponding to the filters in the feature vector is performed, taking into account the calculated significance values of each group of $\bar{\bar{\mathbf{\Phi}}}^h$ reservoir neurons to which the input signals $\mathbf{g}^{CSP}$ are applied. As a result, a vector

$$\mathbf{S}^{SL} = \frac{1}{M_{in}}|\mathbf{S}^{in}|\bar{\bar{\mathbf{\Phi}}}^h = \frac{1}{M_{in}nM_{out}}|\mathbf{S}^{in}|\tilde{\mathbf{G}}|\mathbf{S}^{out}|\mathbf{\Phi} \tag{7}$$

is formed that estimates the contribution of each input signal $\mathbf{g}$ to the classification result.

In determining the significant channels of multivariate fMRI data, the algorithm we developed to identify the most significant input data was repeated 10 times on different cross-validation samples. All calculated significance coefficients were then averaged, resulting in a final estimate of $\bar{\mathbf{S}}^{SL}$ for each channel.

The most significant areas in MDD and ASD classifications are presented in Fig. 4 and Fig. 5, respectively.

## B    HYPERPARAMETER OPTIMIZATION

Let us consider the effect of the pipeline hyperparameters on the classification accuracy presented in Fig. 6.

Fig. 6(A) illustrates the dependence of the best classification accuracy averaged over random partitions of subjects, and the 95% confidence interval as a function of the number of output filter components. Dependence is characterised by a smooth increase in accuracy from 0.74 to 0.87 as the number of components increases from 2 to 20, with accuracy almost ceasing to change when approaching 10.

As can be seen in Fig. 6(B), as the number of components increases from 3 to 14, the classification accuracy increases from 0.76 to 0.87 after which it starts to decrease slightly. Thus, regardless of the number of output components of the CSP filters. As can be seen, a large number of components of both input and output filters should be used to achieve the best classification results.

Fig. 6(C) shows the dependence of the average accuracy value on the leakage rate $\gamma$. Decreasing the value of the leakage parameter $\gamma$ leads to an increase in classification accuracy, and the optimal accuracy is reached at $\gamma \approx 0.1$. This characteristic change in the dependence and the significant decrease in the optimal value of $\gamma$ indicate that reservoirs with larger memory are most suitable for our approach, which allows a more efficient account of temporal dynamics.

This effect is explained by pre-processing the data using CSP filters before feeding it into the reservoir. This process removes redundant spatial information from the model input and extracts key components, each of which is a linear combination of the original fMRI signals. This provides the reservoir with a more compact representation of the data containing only the most relevant spatial and temporal features. As a result, the reservoir does not need to account for unnecessary spatial variations and the focus shifts to analysing temporal dynamics. This allows for the use of lower $\gamma$ values, which increases model memory and improves the processing of temporal patterns. This allows the reservoir to analyse temporal patterns of signals more efficiently.

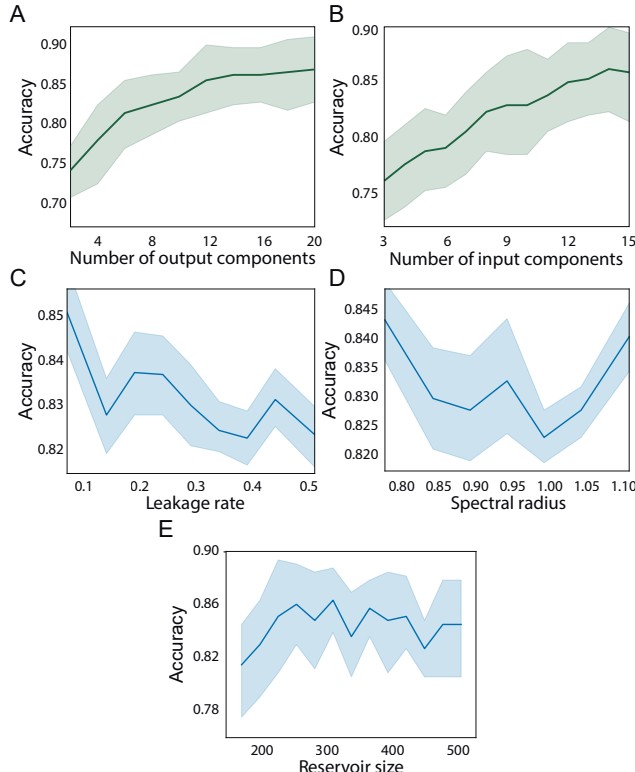

Figure 6: **Dependencies of classification accuracy on different hyperparameters.** Panel (A) illustrates the best results for different number of components of the output CSP filter. The sets of reservoir parameters that provide the best average partitioned accuracy for a given number of output components were selected. Panel (B) shows the dependence of the average accuracy on the number of input CSP-filter components, assuming the other hyperparameters of the model, including the number of output CSP-filter components, are optimal. Panel (C) shows dependency of mean classification accuracy on the leakage rate $\gamma$. Panel (D) shows dependency of mean classification accuracy on the spectral radius $\rho$. Panel (E) shows dependency of mean classification accuracy on the number of reservoir neurons $N_h$. The solid line shows the mean accuracy and the semi-transparent line shows the confidence interval.

Panel (D) in Fig. 6 illustrates the dependency of average classification accuracy on spectral radius $\rho$. Optimal accuracy is achieved at $\rho = 1$, decreasing or increasing $\rho$ yields no performance gains.

Finally, we tested model accuracy dependence on number of reservoir neurons $N_h \in [182, 504]$ with step of 28 neurons (Fig. 6). Accuracy gradually increase as we add more nodes to the reservoir network up to optimal $N_h = 308$, after that addition of neurons does not significantly affect pipeline performance. Based on this result, we fixed $N_h$ at 308.

## C   BENCHMARK METHODS FOR CLASSIFICATION

### C.1   CONSTRUCTION OF FUNCTIONAL CONNECTIVITY MATRICES

Following the traditional approaches for classification biomedical data (Xia et al., 2022; Song et al., 2017; Kurkin et al., 2025), we first construct the functional connectivity matrices by calculating Pearson correlation coefficients for all pairs of BOLD signals for each subject:

$$r_{ij} = \frac{\sum_{t=1}^{T}(X_{it} - \bar{X}_i)(X_{jt} - \bar{X}_j)}{\sqrt{\sum_{t=1}^{T}(X_{it} - \bar{X}_i)^2}\sqrt{\sum_{t=1}^{T}(X_{jt} - \bar{X}_j)^2}}, \tag{8}$$

where $\bar{X}$ is the mean of the $X$ time-series.

We consider only connections with $p$-value $< 0.05$. Due to correlation coefficient varies from -1 to 1, we calculate its absolute value. So, we obtain 163 matrices of $139{\times}139$ size for MDD dataset.

## C.2 NETWORK MEASURES

Each functional connectivity matrix could be represented in the form of a network (graph). To analyze the network's structure and topology, we calculate the following global measures (Andreev et al., 2023; Pisarchik et al., 2023): mean node strength $\langle k^w \rangle$, average shortest path length $\langle L \rangle$, number of edges $N_e$, clustering coefficient $C^{ws}$, and small-world coefficient $\sigma$. Mean node strength is calculated as (Rubinov & Sporns, 2011)

$$\langle k^w \rangle = \frac{1}{N} \sum_{i=1}^{N} k_i^w, \tag{9}$$

where $k_i$ is the strength of i-th node (the sum of weights of edges connected to the node), $N$ is the number of nodes in the graph.

Average shortest path length is calculated as (Van Den Heuvel et al., 2009)

$$\langle L \rangle = \frac{\sum_{i=1}^{N} \sum_{j=1}^{N} L_{ij}}{N(N-1)}, \tag{10}$$

where $L_{ij}$ is the shortest path between $i$-th and $j$-th nodes. Note, that $L_{ii} = 0$ for $i = 1, ..., N$, so we exclude it from calculation.

Clustering coefficient is the Watts-Strogatz clustering coefficient calculated as (Watts & Strogatz, 1998; Costantini & Perugini, 2014)

$$C^{ws} = \frac{1}{N} \sum_{i=1}^{N} 2n_i / k_i^n (k_i^n - 1), \tag{11}$$

where $n_i$ is the number of direct edges interconnecting the $k_i^n$ nearest neighbors of node $i$.

Small-world coefficient is calculated as (Humphries & Gurney, 2008)

$$\sigma = \frac{C^{ws}/C_r}{\langle L \rangle / \langle L_r \rangle}, \tag{12}$$

where $C_r$ and $\langle L_r \rangle$ are the clustering coefficient and the average shortest path length for an Erdős–Rényi random graph with the same number of nodes and edges, respectively.

For calculating the network measures, we utilized the open-source NetworkX package in Python. Those five measures are used as input data for LDA, SVM and kNN.

## C.3 LINEAR DISCRIMINANT ANALYSIS

As a simple benchmark classification method we use Linear Discriminant Analysis (LDA), a supervised machine learning method that allows us to perform dimensionality reduction by projecting the input data to a linear subspace consisting of the directions which maximize the separation between classes (Duda et al., 2006; Mandelkow et al., 2016; Pisarchik et al., 2023). We use a set of the above five network measures as features.

First, we split each group of subjects into train and test subsets in the proportion of 60% by 40%. Then, we apply 100 stratified random permutations for cross-validation, fit the LDA model with the train set, and test it with the test one by calculating the accuracy of the model.

Using LDA, we test different solvers: singular value decomposition (svd), least squares solution (lsqr) and eigenvalue decomposition (eigen). To construct LDA we use the open-source Scikit-Learn package in Python.

### C.4 SUPPORT VECTOR MACHINE

As a second benchmark classification method we use Support Vector Machine (SVM) the most common method for classification fMRI-based data (Bondi et al., 2023; Hui et al., 2024; Grubov et al., 2025; Kuc, 2025). The SVM constructs a maximal margin linear classifier in a high dimensional feature space, by mapping the original features via a kernel function.

Similar to LDA, we use a set of the five network measures as features, split each group of subjects into train and test subsets in the proportion of 60% by 40%, and apply 100 stratified random permutations for cross-validation. For SVM we use RBF-kernel and optimize regularization parameter $C \in (0, 3]$ and kernel coefficient $\delta \in (0, 3]$ using grid search. To construct SVM we use the open-source Scikit-Learn package in Python.

### C.5 K-NEAREST NEIGHBORS

Another effective supervised machine learning algorithm for classification of fMRI data is k-nearest neighbors (kNN) (Misaki et al., 2010; Maher et al., 2023; Minhas et al., 2024; Eser & Erdoğan, 2025). It works by finding the k closest data points (neighbors) to a new, unlabeled data point and then uses those neighbors to predict the class (for classification) or the value (for regression) of the new point.

Using kNN, we apply grid search optimization of the number of neighbors $n_n \in [1, 20]$, power parameter $p \in \{1, 2\}$ and two types of the weight function used in prediction: uniform weights and weight points by the inverse of their distance.

We use the same train and test subsets as for LDA and SVM and apply 100 stratified random permutations for cross-validation. To construct kNN we use the open-source Scikit-Learn package in Python.

### C.6 GRAPH NEURAL NETWORK

As a benchmark deep learning method we choose Graph Neural Network (GNN) which is previously showed good results in classification of fMRI-based data (Morris et al., 2019; Yu et al., 2022; Pitsik et al., 2023).

As the input data for GNN we use the obtained connectivity matrices. As GNN we used a two-layer graph convolutional network with ReLU and sigmoid activation function with the following set of hyperparameters: learning rate of 0.001, 100 epoch, batch size of 20, Adam optimizer, BCE with logits loss function and dropout of 0.2. To provide the stability of the training process, we used stratified 5-fold.

We test all the above models by calculating the accuracy, recall, precision and F1-score of the models.

