# OpenReview forum: "Reservoir Computing with Spatial Filtering and Manifold Learning for fMRI Classification"
_ICLR.cc/2026/Conference — Submitted to ICLR 2026_

### Official Review · Reviewer_o33G · 2025-10-22

**Soundness:** 3
**Presentation:** 3
**Contribution:** 2
**Rating:** 2
**Confidence:** 3

**Summary:**

This paper introduces a pipeline that integrates discriminative spatial filtering with reservoir computing to classify multidimensional spatio-temporal features from high dimensional, multivariate fMRI time series data. The proposed approach applies a supervised spatial filter (CSP) to the reservoir states, thereby enabling the extraction of discriminative spatiotemporal representations. The pipeline also includes a parametric manifold learning via UMAP for nonlinear dimensionality reduction with the final classified carried out by LDA. The authors evaluate the performance of their pipeline on a single HCP dataset for major depressive disorder and claim that their approach outperform LDA, SVM, kNN, and GNN. The paper provides an interpretation of their results on this dataset by consistent identification of cortical and subcortical regions implicated in major depressive disorder.

**Strengths:**

The proposed approach extends the reservoir computing framework by incorporating discriminative spatial filtering and parametric UMAP to enable efficient classification.

There is a significant amount of ablation studies that illustrate the importance of UMAP

**Weaknesses:**

While the end to end pipeline seems to be new,  each of the components in the pipelines has been extensively studied in the literature and the paper lack novelty in this sense.

The empirical evaluation is limited to a single data set.

The computational complexity of CSP is significant, which is applied to the raw data as well as the reservoir states.

Reservoir computing was introduced to reduce the complexity of recurrent neural networks. However currently there are alternatives that
work extremely well in practice e.g. LSTM.

The authors should compare their approach to transformer based approaches that can capture multidimensional spatial and temporal data
The comparison in Table 3 is really unfair and uses different types of inputs for different methods (FC for GNN,  graph metrics for the rest)

**Questions:**

The authors should try to respond to each of the weaknesses mentioned in the previous section.

---

> ### Author Response · Authors · 2025-12-03
> **Response to Reviewer o33G, Part 1**
>
> First of all, we would like to thank the Reviewer for the valuable comments. The Reviewer raised several questions, which we are addressing below. In the revised manuscript the major changes are marked in blue.
>
> &nbsp;
>
> $\textbf{Q1}$: While the end to end pipeline seems to be new, each of the components in the pipelines has been extensively studied in the literature and the paper lack novelty in this sense.
>
> $\textbf{A1}$: While the individual components of our pipeline exist in prior work, the novelty lies in how we integrate them into a reservoir-computing framework that explicitly preserves and leverages the spatial structure of fMRI data. This design is not a simple aggregation of known methods but a spatially informed architecture tailored to BOLD dynamics.
>
> We first apply CSP directly to voxel-wise BOLD signals to obtain supervised spatial filters that emphasize condition-specific variance. Each resulting component is then mapped to a separate group of reservoir neurons, ensuring that the spatial structure learned by CSP is preserved at the moment of entry into the reservoir. The reservoir remains random and recurrent, performing nonlinear temporal encoding without destroying this structure. A second CSP step is applied to the reservoir activations to extract supervised spatial features from the nonlinear embedding. UMAP then provides a compact, geometry-preserving representation that supports both accurate classification and interpretable feature organization.
>
> Ablation analyses show that removing or altering these stages consistently reduces performance, while the full configuration yields the highest accuracy. Moreover, the resulting spatial and temporal patterns align well with established findings in both the MDD dataset and the ABIDE autism dataset, further supporting the validity of the design.
>
> In summary, the contribution is the spatial-structure-preserving reservoir-computing architecture, which combines supervised spatial filtering, structured nonlinear temporal encoding, and geometry feature reduction in a way that has not previously been applied to fMRI and that improves both accuracy and interpretability.
>
> &nbsp;
>
> $\textbf{Q2}$: The empirical evaluation is limited to a single data set.
>
> $\textbf{A2}$: We have tested our approach additionally on the fMRI data from the NYU site of public ABIDE dataset. The achieved results are shown below and added to the new version of the paper:
>
> Metric                 | Accuracy                         | Recall                         | Precision                         | F1-Score                         |
> |---------------------------|-------------------------------|-------------------------------|-------------------------------|-------------------------------|
> | CSP-RC-CSP-UMAP-LDA   | $0.67 \pm 0.07$   | $0.71 \pm 0.10$   | $0.69 \pm 0.05$   | $0.70 \pm 0.07$ |
>
> Comparing with the recent results from [Xue, Yanfang, et al. "Dynamic functional connections analysis with spectral learning for brain disorder detection." Artificial Intelligence in Medicine 157 (2024): 102984] obtained on the same dataset, we can conclude that the accuracy of our approach is comparable with the SOTA methods including transformers: only 3 methods demonstrate higher accuracy. Moreover, F1-score of our approach is higher than almost all of them (only DART achieved 73%). Also, the most significant regions are well known as highly involved in autism according to the literature.
>
>
> &nbsp;
>
> $\textbf{Q3}$: The computational complexity of CSP is significant, which is applied to the raw data as well as the reservoir states.
>
> $\textbf{A3}$:  The computational cost of CSP in our pipeline is very low. CSP operates on covariance matrices and is known to be a fast and lightweight procedure. In our implementation, the cost is further reduced because we retain only a small subset of the most discriminative CSP components at each stage, which substantially reduces the size of the matrices involved and makes both CSP applications extremely efficient. Finally, ablation analyses show that both CSP steps significantly improve discriminability, while their added computational overhead is minimal. Thus, the use of CSP is justified not only conceptually but also computationally

---

> ### Author Response · Authors · 2025-12-03
> **Response to Reviewer o33G, Part 2**
>
> $\textbf{Q4}$: Reservoir computing was introduced to reduce the complexity of recurrent neural networks. However currently there are alternatives that work extremely well in practice e.g. LSTM.
>
> $\textbf{A4}$:  Reservoir computing provides a strong form of architectural regularization, because the recurrent part of the model is fixed and contains no trainable parameters. This prevents overfitting in the recurrent dynamics and is particularly suitable for fMRI data, where sample sizes are small and time series are short.
>
> In contrast, models such as LSTM must optimize all recurrent and input–output weights, resulting in thousands of learnable parameters. Under typical neuroimaging conditions, this substantially increases the risk of overfitting and makes training more sensitive to hyperparameters, initialization, and regularization choices.
>
> Thus, by avoiding the need to train a large recurrent network, reservoir computing offers a more robust and data-efficient alternative for fMRI time-series analysis, while still providing rich nonlinear temporal representations.
>
> &nbsp;
>
> $\textbf{Q5}$: The authors should compare their approach to transformer based approaches that can capture multidimensional spatial and temporal data The comparison in Table 3 is really unfair and uses different types of inputs for different methods (FC for GNN, graph metrics for the rest)
>
> $\textbf{A5}$: We have made the comparison with a plethora of fMRI classification approaches on another open dataset. Comparing with the recent results from [Xue, Yanfang, et al. "Dynamic functional connections analysis with spectral learning for brain disorder detection." Artificial Intelligence in Medicine 157 (2024): 102984] obtained on the same dataset, we can conclude that the accuracy of our approach is comparable with the SOTA methods including transformers: only 3 methods (CRNN, BolT, and dCSL) demonstrate higher accuracy. Moreover, F1-score of our approach is higher than almost all of them (only DART achieved 73%). On the other side, our approach provides interpretability. The most significant regions (channels) identified by our model are consistently reported in the literature as strongly implicated in autism. This correspondence supports the interpretability of the proposed framework and indicates that it achieves reliable classification performance.

---

### Official Review · Reviewer_WaDm · 2025-10-28

**Soundness:** 2
**Presentation:** 2
**Contribution:** 2
**Rating:** 2
**Confidence:** 3

**Summary:**

The paper proposes a classification pipeline for fMRI data by combining spatial filtering (CSP), reservoir computing (RC), UMAP and LDA which it claims gives state of the art accuracy on a MDD vs Controls dataset.

**Strengths:**

The paper is well motivated in the sense that it aims to utilize both spatial and temporal information present in fMRI data to perform classification. It also does a good job of combining existing methods without trying to reinvent any wheels.

**Weaknesses:**

This is largely an application paper, which means it really needs to include robust experiments to prove the claims being made. In my opinion it falls significantly short of that.
1. It claims to be state of the art, but the comparisons are only made against baseline methods, rather than making an honest attempt at comparing against a plethora of fMRI classification approaches from the literature.
2. Experiments are provided only on a single dataset, and since it's unclear whether it's publicly available or not, it's hard to know how much work has already been done on that particular dataset. Either way, showing a method works on a single dataset is simply not enough.
3. train/test split. It's unclear what was used for hyperparameter tuning, was there a completely blind test set that was set apart to evaluate performance? or is everything reported here simply on train/test splits with no separation between validation and testing.
4. generalization. It's unclear how well does this approach generalize. Again, it's impossible to establish anything beyond cross subject generalization here. If the method is truly being presented as "SOTA" for MDD classification it needs to be shown that it can generalize out of dataset / different sites etc.
5. ablation. If a pipeline is being proposed, it'd make sense to ablate different components of the pipeline to show how they contribute (just like the paper does for UMAP).

overall writing of the paper can also be improved. please use parentheses for citation and also use consistent notation throughout the paper.

**Questions:**

see weaknesses.

---

> ### Author Response · Authors · 2025-12-03
> **Response to Reviewer WaDm, Part 1**
>
> First of all, we would like to thank the Reviewer for the valuable comments. The Reviewer raised several questions, which we are addressing below. In the revised manuscript the major changes are marked in blue.
>
> &nbsp;
>
> $\textbf{Q1}$: It claims to be state of the art, but the comparisons are only made against baseline methods, rather than making an honest attempt at comparing against a plethora of fMRI classification approaches from the literature.
>
> $\textbf{A1}$: We have made the comparison with a plethora of fMRI classification approaches on another open dataset. Comparing with the recent results from [Xue, Yanfang, et al. "Dynamic functional connections analysis with spectral learning for brain disorder detection." Artificial Intelligence in Medicine 157 (2024): 102984] obtained on the same dataset, we can conclude that the accuracy of our approach is comparable with the SOTA methods including transformers: only 3 methods demonstrate higher accuracy. Moreover, F1-score of our approach is higher than almost all of them (only DART achieved 73%).
>
> On the other side, our approach provides interpretability. We have identified 9 fMRI regions with significant differences between ASD and HC groups. Notably, 5 of these regions are located in the cerebellum, one of the most consistent sites of abnormality in ASD patients [D’Mello, A.M., Stoodley, C.J., 2015. Cerebro-cerebellar circuits in autism spectrum disorder. Front. Neurosci. 9]. Four of them constitute right and left cerebellar Crus I/II, a region shown to have reduced gray matter volume and decreased fractional anisotropy in white matter tracks connecting this region to the dentate nucleus and contralateral cerebral cortex, structural deficits correlated with impaired communication, social interaction deficits, and repetitive behaviors [D’Mello, A.M., Stoodley, C.J., 2015. Cerebro-cerebellar circuits in autism spectrum disorder. Front. Neurosci. 9]. The fifth significant cerebellum region is lobule X, shown to have hyper-connectivity in ASD patients, which may contribute to atypical visual exploration and eye movement control difficulties [Lanciano, T., Petri, G., Gili, T., Bonchi, F., 2025. Contrast subgraphs catch patterns of altered functional connectivity in autism spectrum disorder. Sci Rep 15, 24265].
>
> Three other significant regions belong to the orbitofrontal cortex. Laminar-specific imbalances in excitatory and inhibitory circuits in the orbitofrontal cortex are observed in ASD patients. Myelinated axons, proxies for excitatory pathways, show reduced density and diameter across layers in ASD, alongside lower excitatory neuron density. These changes likely disrupt OFC communications with limbic cortices and the amygdala, providing an anatomic basis for social interaction and emotional deficits in ASD [Liu, X., Bautista, J., Liu, E., Zikopoulos, B., 2020. Imbalance of laminar-specific excitatory and inhibitory circuits of the orbitofrontal cortex in autism. Molecular Autism 11, 83].
>
> The last detected channel with significant difference between HC and ASD is the right middle temporal pole, a region where increased cortical thickness was associated with more severe communication impairment in ASD patients [Pereira, A.M. et al. 2018. Differences in Cortical Structure and Functional MRI Connectivity in High Functioning Autism. Front. Neurol. 9].
>
> So, we can conclude that the proposed approach is a powerful interpretable tool which also provides good accuracy of classification.

---

> ### Author Response · Authors · 2025-12-03
> **Response to Reviewer WaDm, Part 2**
>
> $\textbf{Q2}$: Experiments are provided only on a single dataset, and since it's unclear whether it's publicly available or not, it's hard to know how much work has already been done on that particular dataset. Either way, showing a method works on a single dataset is simply not enough.
>
> $\textbf{A2}$: We have tested our approach additionally on the fMRI data from the NYU site of public ABIDE dataset which is often used as a benchmark in similar studies [Heinsfeld, Anibal Sólon, et al. "Identification of autism spectrum disorder using deep learning and the ABIDE dataset." NeuroImage: clinical 17 (2018): 16-23; Xue, Yanfang, et al. "Dynamic functional connections analysis with spectral learning for brain disorder detection." Artificial Intelligence in Medicine 157 (2024): 102984; Xiao, Lei, et al. "Classification of autism spectrum disorders based on the adaptive fuzzy reasoning system." Biomedical Signal Processing and Control 104 (2025): 107513; Nogay, Hidir Selcuk, and Hojjat Adeli. "Multiple classification of brain MRI autism spectrum disorder by age and gender using deep learning." Journal of Medical Systems 48.1 (2024): 15]. The achieved results are shown below and added to the new version of the paper:
>
> Metric                 | Accuracy                         | Recall                         | Precision                         | F1-Score                         |
> |---------------------------|-------------------------------|-------------------------------|-------------------------------|-------------------------------|
> | CSP-RC-CSP-UMAP-LDA   | $0.67 \pm 0.07$   | $0.71 \pm 0.10$   | $0.69 \pm 0.05$   | $0.70 \pm 0.07$ |
>
> Comparing with the recent results from [Xue, Yanfang, et al. "Dynamic functional connections analysis with spectral learning for brain disorder detection." Artificial Intelligence in Medicine 157 (2024): 102984] obtained on the same dataset, we can conclude that the accuracy of our approach is comparable with the most modern ML methods including transformers: only 3 methods demonstrate higher accuracy. Moreover, F1-score of our approach is higher than almost all of them (only DART achieved 73%).
>
> Also, the most significant regions are well known as highly involved in autism according to the literature. We have identified 9 fMRI regions with significant differences between ASD and HC groups. Notably, 5 of these regions are located in the cerebellum, one of the most consistent sites of abnormality in ASD patients [D’Mello, A.M., Stoodley, C.J., 2015. Cerebro-cerebellar circuits in autism spectrum disorder. Front. Neurosci. 9]. Three other significant regions belong to the orbitofrontal cortex. Laminar-specific imbalances in excitatory and inhibitory circuits in the orbitofrontal cortex are observed in ASD patients [Liu, X., Bautista, J., Liu, E., Zikopoulos, B., 2020. Imbalance of laminar-specific excitatory and inhibitory circuits of the orbitofrontal cortex in autism. Molecular Autism 11, 83]. The last detected channel with significant difference between HC and ASD is the right middle temporal pole, a region where increased cortical thickness was associated with more severe communication impairment in ASD patients [Pereira, A.M. et al.  2018. Differences in Cortical Structure and Functional MRI Connectivity in High Functioning Autism. Front. Neurol. 9].
>
> &nbsp;
>
> $\textbf{Q3}$: train/test split. It's unclear what was used for hyperparameter tuning, was there a completely blind test set that was set apart to evaluate performance? or is everything reported here simply on train/test splits with no separation between validation and testing.
>
> $\textbf{A3}$: We employed a 10-fold stratified shuffle-split cross-validation for both hyperparameter tuning and performance evaluation. We used this method due to limited amount of available data (setting aside a completely blind test set would significantly decrease valuable training data).
>
> &nbsp;
>
> $\textbf{Q4}$: generalization. It's unclear how well does this approach generalize. Again, it's impossible to establish anything beyond cross subject generalization here. If the method is truly being presented as "SOTA" for MDD classification it needs to be shown that it can generalize out of dataset / different sites etc.
>
> $\textbf{A4}$: We have tested our approach additionally on the fMRI data from public dataset which contains healthy and autism subjects. It shown good accuracy results with correct interpretability. So, our approach can work well with different diseases and datasets.

---

> ### Author Response · Authors · 2025-12-03
> **Response to Reviewer WaDm, Part 3**
>
> $\textbf{Q5}$: ablation. If a pipeline is being proposed, it'd make sense to ablate different components of the pipeline to show how they contribute (just like the paper does for UMAP).
>
> $\textbf{A5}$: We have done additional calculations for different ablation schemes. The results are shown below and added to the new version of the paper:
>
> | Metric                 | Accuracy                         | Recall                         | Precision                         | F1-Score                         |
> |---------------------------|-------------------------------|-------------------------------|-------------------------------|-------------------------------|
> | CSP-RC-CSP-UMAP-LDA   | $\\mathbf{0.87 \\pm 0.05}$    | $\\mathbf{0.80 \\pm 0.10}$    | $\\mathbf{0.88 \\pm 0.09}$    | $\\mathbf{0.83 \\pm 0.07}$    |
> | CSP-RC-UMAP-LDA           | $0.86 \\pm 0.06$              | $0.78 \\pm 0.11$              | $0.85 \\pm 0.08$              | $0.81 \\pm 0.08$              |
> | CSP-RC-CSP-LDA            | $0.85 \\pm 0.05$              | $0.72 \\pm 0.10$              | $0.87 \\pm 0.07$              | $0.79 \\pm 0.08$              |
> | RC-UMAP-LDA               | $0.59 \\pm 0.06$              | $0.20 \\pm 0.15$              | $0.44 \\pm 0.17$              | $0.25 \\pm 0.16$              |
> | RC-CSP-UMAP-LDA           | $0.57 \\pm 0.10$              | $0.16 \\pm 0.12$              | $0.44 \\pm 0.33$              | $0.22 \\pm 0.16$              |
>
> As one can see, the double-CSP+UMAP architecture provides the highest accuracy while all others are lower. The schemes without either CSP-2 or UMAP also provide high accuracy. The schemes without input CSP cannot achieve good classification results.

---

### Official Review · Reviewer_tTvF · 2025-10-31

**Soundness:** 3
**Presentation:** 2
**Contribution:** 2
**Rating:** 4
**Confidence:** 3

**Summary:**

The paper presents a multi-stage pipeline for classifying resting-state fMRI data from individuals with major depressive disorder (MDD) and healthy controls. The approach combines Common Spatial Patterns (CSP) for spatial filtering, Reservoir Computing (RC) for temporal encoding, parametric UMAP for nonlinear dimensionality reduction, and LDA for classification. The authors report an accuracy of 87% (F1 score of 0.83) on a dataset of 163 participants, outperforming basic baselines including SVMs, kNNs, and GNNs trained on functional connectivity matrices. They also provide region-level interpretability analysis via SHAP value back-projection to brain regions.

**Strengths:**

- Addresses a relevant and interdisciplinary problem (neuroimaging classification with limited data).
- The method is computationally simple and interpretable, combining established signal processing and manifold learning steps.
- Experimental setup and preprocessing are clearly described; evaluation uses subject-wise cross-validation.
- The reported classification accuracy is relatively high for the given dataset.

**Weaknesses:**

1. Not ICLR-level novelty.
The paper essentially stacks well-known methods (CSP, RC, UMAP, LDA) without demonstrating new theoretical or algorithmic insights. The main idea—applying CSP on reservoir states—is incremental and not well justified conceptually.

2. Pipeline without integration.
The approach feels like a sequence of independent components rather than a coherent model. There is no end-to-end learning (as far as I understand) or extended ablation showing why each component is necessary.

3. Limited baselines and older references.
The comparison set is narrow and dated (LDA, SVM, kNN, GNN on connectivity). More competitive baselines such as temporal CNNs, GRU-ODE-Bayes, transformers, or contrastive representation learning approaches are missing.

4. Small dataset and potential overfitting.
With only 163 subjects, high reported accuracy (87%) may overstate generalization. The evaluation relies solely on within-dataset cross-validation without a separate held-out or external validation set.

5. Unclear contribution to machine learning.
While the application is interesting, the work reads more like a domain-specific methods report than an ML conference paper. There is no theoretical contribution or clear takeaway for the broader ML community.

6. Writing and framing.
The manuscript is verbose and reads more like a research report than a conference submission. Some methodological descriptions (e.g., CSP, LDA equations) occupy space but add little conceptual clarity.

**Questions:**

- Can the authors confirm that the parametric UMAP was fit exclusively on the training folds in each cross-validation iteration (to avoid information leakage to the test set)?
- What is the variance across folds—does performance generalize beyond 163 subjects?
- How do your results compare to recent deep representation learning approaches for fMRI (e.g., BrainLM, transformers, or graph contrastive learning)?

---

> ### Author Response · Authors · 2025-12-03
> **Response to Reviewer tTvF**
>
> First of all, we would like to thank the Reviewer for the valuable comments. The Reviewer raised several questions, which we are addressing below. In the revised manuscript the major changes are marked in blue.
>
> &nbsp;
>
> $\textbf{Q1}$:Can the authors confirm that the parametric UMAP was fit exclusively on the training folds in each cross-validation iteration (to avoid information leakage to the test set)?
>
> $\textbf{A1}$: All CSP filters, parametric UMAP, and LDA are fitted only on data of subjects from training folds in each cross-validation iteration, so data leakage is avoided.
>
> &nbsp;
>
> $\textbf{Q2}$: What is the variance across folds—does performance generalize beyond 163 subjects?
>
> $\textbf{A2}$: We explicitly report the standard deviation across all test folds in Table 3: 87% ± 5% for our full pipeline. This relatively low variance indicates stable performance across different random splits of the 163-subject dataset. The ±5% range reflects the inherent heterogeneity of MDD rather than model instability. To test how our method performance generalize beyond these 163 subjects, we also tested our approach on the fMRI data from the NYU site of public ABIDE dataset to classify ASD patients. The achieved results are shown below and added to the new version of the paper:
>
> Metric                 | Accuracy                         | Recall                         | Precision                         | F1-Score                         |
> |---------------------------|-------------------------------|-------------------------------|-------------------------------|-------------------------------|
> | CSP-RC-CSP-UMAP-LDA   | $0.67 \pm 0.07$   | $0.71 \pm 0.10$   | $0.69 \pm 0.05$   | $0.70 \pm 0.07$ |
>
> Achieved accuracy is lower, than on the original dataset, but, comparing these results with models from [Xue, Yanfang, et al. "Dynamic functional connections analysis with spectral learning for brain disorder detection." Artificial Intelligence in Medicine 157 (2024): 102984] obtained on the same ABIDE dataset, we can conclude that the accuracy of our approach is comparable with SOTA methods including transformers: only 3 methods demonstrate higher accuracy. Moreover, F1-score of our approach is higher than almost all of them (only DART achieved 73%).
>
> &nbsp;
>
> $\textbf{Q3}$: How do your results compare to recent deep representation learning approaches for fMRI (e.g., BrainLM, transformers, or graph contrastive learning)?
>
> $\textbf{A3}$: Our method outperforms recent deep representation learning approaches on the fMRI classification task in terms of F1-score. Specifically, on the publicly available NYU site of the ABIDE dataset, our framework achieves an F1-score of 0.70. In comparison, Xue et al. (2024) reported an F1-score of 0.66 using a blood-oxygen-level-dependent (BOLD) transformer, which is a state-of-the-art deep learning model designed for dynamic functional connectivity analysis. This indicates that our integrated approach, combining reservoir computing, supervised spatial filtering, and parametric UMAP, not only matches but exceeds the performance of contemporary transformer-based models in discriminating between clinical groups based on resting-state fMRI data. The results of application graph neural networks and different deep-learning nethods for fMRI-data classification are also lower than our approach demonstrates: in [Xue et al. (2024)] BrainGNN demonstrates F1 = 0.57, MVS-GCN – 0.49, MDGL – 0.58, BrainTGL – 0.43, BrainGB – 0.55. Moreover, our approach provides interpretability. We have identified 9 fMRI regions with significant differences between ASD and HC groups which are well known as highly involved in autism according to the literature. We have identified 9 fMRI regions with significant differences between ASD and HC groups. Notably, 5 of these regions are located in the cerebellum, one of the most consistent sites of abnormality in ASD patients [D’Mello, A.M., Stoodley, C.J., 2015. Cerebro-cerebellar circuits in autism spectrum disorder. Front. Neurosci. 9]. Three other significant regions belong to the orbitofrontal cortex. Laminar-specific imbalances in excitatory and inhibitory circuits in the orbitofrontal cortex are observed in ASD patients [Liu, X. et al. 2020. Imbalance of laminar-specific excitatory and inhibitory circuits of the orbitofrontal cortex in autism. Molecular Autism 11, 83]. The last detected channel with significant difference between HC and ASD is the right middle temporal pole, a region where increased cortical thickness was associated with more severe communication impairment in ASD patients [Pereira, A.M. et al. 2018. Differences in Cortical Structure and Functional MRI Connectivity in High Functioning Autism. Front. Neurol. 9].

---

### Official Review · Reviewer_2U6W · 2025-10-31

**Soundness:** 2
**Presentation:** 1
**Contribution:** 1
**Rating:** 2
**Confidence:** 4

**Summary:**

This paper proposes a multi-stage pipeline combining Common Spatial Patterns (CSP), Reservoir Computing (RC), and parametric UMAP for classifying resting-state fMRI data from Major Depressive Disorder (MDD) patients vs. healthy controls. The approach applies CSP twice (before and after RC) to extract spatial features, uses RC for temporal encoding, applies UMAP for dimensionality reduction, and performs LDA classification. On 163 participants, the method achieves 87% accuracy, outperforming classical baselines. While the problem is clinically relevant and the interpretability analysis provides biological insights, the work suffers from limited novelty, insufficient theoretical justification, questionable domain transfer from EEG to fMRI, and weak experimental validation.

**Strengths:**

- **Clinically important problem**: MDD classification from resting-state fMRI addresses a significant challenge in computational psychiatry with potential real-world impact.
- **Interpretability**: The backward propagation of SHAP values to identify relevant brain regions (Fig. 3) provides biologically plausible explanations, highlighting areas like the medial superior frontal gyrus (DMN) that are known to be associated with MDD.
- **End-to-end framework**: The paper presents a complete pipeline from raw fMRI signals to classification with detailed implementation.
- **Comprehensive baseline comparison**: The authors compare against multiple classical methods (LDA, SVM, kNN, GNN) on the same dataset.

**Weaknesses:**

1. **Limited novelty**:
    - The paper combines existing techniques (CSP from 2000, RC from 2000s, UMAP from 2018) without substantial innovation
    - The main claimed contribution is "applying CSP to reservoir states rather than raw data" - this is an engineering choice, not a methodological advance
    - Even the input matrix G structure is borrowed from Hramov et al. (2024)
    - **For ICLR 2026, this level of novelty is insufficient**
2. **Lack of theoretical justification**:
    - **Critical**: Why should this specific combination of methods work? The paper provides no principled explanation
    - Why apply CSP twice? What is the theoretical motivation for spatial filtering → temporal encoding → spatial filtering again?
    - Why is UMAP necessary after CSP produces discriminative features?
    - The design appears entirely empirical ("we tried this and it worked") without understanding *why*
3. **Questionable domain transfer (EEG → fMRI)**:
    - CSP was designed for EEG/BCI applications with fundamentally different signal characteristics
    - **EEG**: high temporal resolution (~1000 Hz), low spatial resolution, electrical activity
    - **fMRI**: low temporal resolution (~0.5 Hz), better spatial resolution, hemodynamic response
    - **The paper provides no justification for why an EEG-based spatial filtering method should be appropriate for fMRI's slow hemodynamic signals**
    - All Related Works references (Section 2.2) discuss EEG applications, yet the method is applied to fMRI without domain-specific adaptation
4. **Small dataset and overfitting concerns**:
    - 163 subjects is small for neuroimaging in 2024-2025 (typical studies use 500-5000+ subjects)
    - 87% ± 0.05 accuracy on such a small dataset raises serious overfitting concerns
    - No validation on independent datasets or multi-site data
    - Suggest: Test on public datasets (ABIDE, UK Biobank, HCP) to demonstrate generalization
5. **Insufficient ablation studies**:
    - Table 3 only shows: full pipeline, without UMAP, and without RC
    - **Missing critical ablations**:
        - CSP-I only (no RC, no CSP-II)
        - RC only (no CSP at all)
        - CSP-II only (no CSP-I)
        - Single CSP vs. double CSP comparison
    - Without systematic ablation, we cannot assess each component's contribution
6. **Weak baselines**:
    - Classical methods (LDA, SVM, kNN on graph metrics) are outdated for 2025
    - 2-layer GNN achieving only 64% suggests implementation issues
    - **Missing**: Recent deep learning methods (BrainNetCNN 2017+, Transformer-based models 2020+, state-of-the-art fMRI classification methods from 2023-2024)
7. **Misleading terminology**:
    - The paper claims "explicit spatial modeling" but CSP only maximizes variance ratios, not spatial topology
    - Calling this "spatial optimization" implies learnable parameters, but CSP is a closed-form eigendecomposition

**Questions:**

1. **What is the theoretical justification for applying EEG-based CSP to fMRI?** Given the fundamental differences in signal characteristics (temporal resolution, SNR, physiological basis), why should variance-based spatial filtering designed for EEG work for hemodynamic responses?
2. **Why is the double-CSP architecture necessary?** Please provide ablation studies comparing:
    - Raw → RC → UMAP → LDA
    - CSP-I → RC → UMAP → LDA
    - RC → CSP-II → UMAP → LDA
    - CSP-I → RC → CSP-II → LDA (no UMAP)
3. **What is the actual novelty claim?** If all components (CSP, RC with separated inputs, UMAP, LDA) are from prior work, what is the core contribution beyond empirical combination?
4. **How do you address overfitting concerns?** With 163 subjects and 87% accuracy, what evidence supports generalization? Can you test on independent public datasets?
5. **Why does the baseline GNN only achieve 64%?** This is surprisingly low for graph-based fMRI classification. Is there an implementation issue, or does this suggest the dataset has low signal?
6. Figure 2: What are the axes for the distribution plots? How many reservoir configurations were tested in total?
7. Why use parametric UMAP instead of simpler alternatives like PCA or kernel PCA? What is lost if you use linear dimensionality reduction?
8. How sensitive is the pipeline to hyperparameters? Varying Minp, Mout, reservoir size, etc.?

---

> ### Author Response · Authors · 2025-12-03
> **Response to Reviewer 2U6W, Part 1**
>
> First of all, we would like to thank the Reviewer for the valuable comments. The Reviewer raised several questions, which we are addressing below. In the revised manuscript the major changes are marked in blue.
>
> $\textbf{Q1}$: What is the theoretical justification for applying EEG-based CSP to fMRI? Given the fundamental differences in signal characteristics (temporal resolution, SNR, physiological basis), why should variance-based spatial filtering designed for EEG work for hemodynamic responses?
>
> $\textbf{A1}$: We apply CSP to fMRI not as an EEG-specific technique but as a general supervised spatial filtering method that operates on covariance structure. CSP requires only multichannel measurements and condition-dependent covariance differences, and therefore does not rely on electrophysiological assumptions. These requirements are directly met in fMRI, where each voxel (or ROI) represents a channel and BOLD signals exhibit substantial spatial mixing and correlated fluctuations due to the hemodynamic point-spread function and physiological noise. This spatial covariance structure in BOLD data has been well documented [Triantafyllou, Christina, et al. "Comparison of physiological noise at 1.5 T, 3 T and 7 T and optimization of fMRI acquisition parameters." Neuroimage 26.1 (2005): 243-250. ], providing a clear rationale for applying variance-based spatial filters.
>
> Taken together, the intrinsic spatial covariance properties of BOLD signals provide a clear rationale for using variance-based spatial filters in fMRI. Accordingly, in our framework CSP is employed strictly as a supervised spatial filtering mechanism that enhances condition-specific BOLD variance, without relying on EEG-specific biophysical assumptions.
>
> &nbsp;
>
> $\textbf{Q2}$: Why is the double-CSP architecture necessary? Please provide ablation studies comparing:
>
> o	Raw → RC → UMAP → LDA
>
> o	CSP-I → RC → UMAP → LDA
>
> o	RC → CSP-II → UMAP → LDA
>
> o	CSP-I → RC → CSP-II → LDA (no UMAP)
>
> $\textbf{A2}$: We have done additional calculations for different ablation schemes. The results are shown below and added to the new version of the paper:
>
> | Metric                 | Accuracy                         | Recall                         | Precision                         | F1-Score                         |
> |---------------------------|-------------------------------|-------------------------------|-------------------------------|-------------------------------|
> | CSP-RC-CSP-UMAP-LDA   | $\\mathbf{0.87 \\pm 0.05}$    | $\\mathbf{0.80 \\pm 0.10}$    | $\\mathbf{0.88 \\pm 0.09}$    | $\\mathbf{0.83 \\pm 0.07}$    |
> | CSP-RC-UMAP-LDA           | $0.86 \\pm 0.06$              | $0.78 \\pm 0.11$              | $0.85 \\pm 0.08$              | $0.81 \\pm 0.08$              |
> | CSP-RC-CSP-LDA            | $0.85 \\pm 0.05$              | $0.72 \\pm 0.10$              | $0.87 \\pm 0.07$              | $0.79 \\pm 0.08$              |
> | RC-UMAP-LDA               | $0.59 \\pm 0.06$              | $0.20 \\pm 0.15$              | $0.44 \\pm 0.17$              | $0.25 \\pm 0.16$              |
> | RC-CSP-UMAP-LDA           | $0.57 \\pm 0.10$              | $0.16 \\pm 0.12$              | $0.44 \\pm 0.33$              | $0.22 \\pm 0.16$              |
>
> As one can see, the double-CSP+UMAP architecture provides the highest accuracy while all others are lower. The schemes without either CSP-2 or UMAP also provide high but lower accuracy. Also, the schemes without input CSP cannot achieve good classification results.

---

> ### Author Response · Authors · 2025-12-03
> **Response to Reviewer 2U6W, Part 2**
>
> $\textbf{Q3}$: What is the actual novelty claim? If all components (CSP, RC with separated inputs, UMAP, LDA) are from prior work, what is the core contribution beyond empirical combination?
>
> $\textbf{A3}$:  Our core contribution is a novel spatially structured reservoir computing architecture explicitly designed for fMRI, which systematically preserves and exploits the spatial discriminability of BOLD signals throughout the temporal encoding process. While individual components (CSP, RC, UMAP, LDA) are known, their integration here is not a simple ensemble, but a principled pipeline that addresses a key gap in fMRI analysis: maintaining spatially informative features across nonlinear temporal transformations.
>
> The novelty lies in three design innovations:
> 1) Spatially Separated Reservoir Inputs.
> Each CSP-derived spatial component is fed into a distinct reservoir neuron group, preventing the blending of discriminative spatial patterns at the input stage—a departure from standard RC where all inputs are fully mixed.
>
> 2) Dual-Stage Supervised Filtering.
> We apply CSP twice: first to raw BOLD signals to extract maximally discriminative spatial patterns, and second to reservoir states to extract discriminative temporal dynamics from the nonlinearly transformed space. This dual filtering ensures that both spatial and temporal discriminability are supervisedly optimized.
>
> 3) Structure-Preserving Pipeline.
> The architecture enforces a spatial -> temporal -> spatial -> geometric flow, ensuring that the spatial organization of fMRI data informs every stage: from input structuring through nonlinear dynamics to final visualization and classification.
> This design is empirically validated by ablation studies, where removing any stage reduces performance, and by cross-dataset generalization (MDD and ABIDE). Moreover, our model retains interpretability, revealing brain regions consistent with established neuropathology—a key advantage over black-box alternatives.
>
> Thus, our contribution is a novel fMRI-specific architecture that integrates known tools in a previously unexplored, spatially aware configuration, improving both accuracy and interpretability for clinical fMRI classification.

---

> ### Author Response · Authors · 2025-12-03
> **Response to Reviewer 2U6W, Part 3**
>
> $\textbf{Q4}$: How do you address overfitting concerns? With 163 subjects and 87% accuracy, what evidence supports generalization? Can you test on independent public datasets?
>
> $\textbf{A4}$: We have tested our approach additionally on the fMRI data from the NYU site of public ABIDE dataset. The achieved results are shown below and added to the new version of the paper:
>
> Metric                 | Accuracy                         | Recall                         | Precision                         | F1-Score                         |
> |---------------------------|-------------------------------|-------------------------------|-------------------------------|-------------------------------|
> | CSP-RC-CSP-UMAP-LDA   | $0.67 \pm 0.07$   | $0.71 \pm 0.10$   | $0.69 \pm 0.05$   | $0.70 \pm 0.07$ |
>
> Comparing with the recent results from [Xue, Yanfang, et al. "Dynamic functional connections analysis with spectral learning for brain disorder detection." Artificial Intelligence in Medicine 157 (2024): 102984] obtained on the same dataset, we can conclude that the accuracy of our approach is comparable with the SOTA methods including transformers: only 3 methods demonstrate higher accuracy. Moreover, F1-score of our approach is higher than almost all of them (only DART achieved 73%).
>
> On the other side, our approach provides interpretability. We have identified 9 fMRI regions with significant differences between ASD and HC groups. Notably, 5 of these regions are located in the cerebellum, one of the most consistent sites of abnormality in ASD patients [D’Mello, A.M., Stoodley, C.J., 2015. Cerebro-cerebellar circuits in autism spectrum disorder. Front. Neurosci. 9]. Four of them constitute right and left cerebellar Crus I/II, a region shown to have reduced gray matter volume and decreased fractional anisotropy in white matter tracks connecting this region to the dentate nucleus and contralateral cerebral cortex, structural deficits correlated with impaired communication, social interaction deficits, and repetitive behaviors [D’Mello, A.M., Stoodley, C.J., 2015. Cerebro-cerebellar circuits in autism spectrum disorder. Front. Neurosci. 9]. The fifth significant cerebellum region is lobule X, shown to have hyper-connectivity in ASD patients, which may contribute to atypical visual exploration and eye movement control difficulties [Lanciano, T., Petri, G., Gili, T., Bonchi, F., 2025. Contrast subgraphs catch patterns of altered functional connectivity in autism spectrum disorder. Sci Rep 15, 24265].
>
> Three other significant regions belong to the orbitofrontal cortex. Laminar-specific imbalances in excitatory and inhibitory circuits in the orbitofrontal cortex are observed in ASD patients. Myelinated axons, proxies for excitatory pathways, show reduced density and diameter across layers in ASD, alongside lower excitatory neuron density. These changes likely disrupt OFC communications with limbic cortices and the amygdala, providing an anatomic basis for social interaction and emotional deficits in ASD [Liu, X., Bautista, J., Liu, E., Zikopoulos, B., 2020. Imbalance of laminar-specific excitatory and inhibitory circuits of the orbitofrontal cortex in autism. Molecular Autism 11, 83].
>
> The last detected channel with significant difference between HC and ASD is the right middle temporal pole, a region where increased cortical thickness was associated with more severe communication impairment in ASD patients [Pereira, A.M., et al. 2018. Differences in Cortical Structure and Functional MRI Connectivity in High Functioning Autism. Front. Neurol. 9].
>
> So, we can conclude that the proposed approach is a powerful interpretable tool which also provides good accuracy of classification.
>
> &nbsp;
>
> $\textbf{Q5}$: Why does the baseline GNN only achieve 64%? This is surprisingly low for graph-based fMRI classification. Is there an implementation issue, or does this suggest the dataset has low signal?
>
> $\textbf{A5}$: First of all, we would note that GNN not always demonstrate high classification results. For instance, in work [Xue Y. et al. "Dynamic functional connections analysis with spectral learning for brain disorder detection." Artificial Intelligence in Medicine 157 (2024): 102984] GNN-based method achieved the lowest accuracy (56%) among the others. Second, investigated dataset is not big enough, and it is known that DL-methods in such case mostly cannot be generalized well [Zhang C. et al. "Understanding deep learning requires rethinking generalization." ICLR. 2017].  Moreover, MDD is known for its heterogeneous neural signatures, which may not be fully captured by static functional connectivity graphs alone. Our higher performance with the proposed RC-based pipeline suggests that preserving spatial structure and leveraging temporal dynamics, aspects not emphasized in the GNN implementation, are critical for this task.

---

> ### Author Response · Authors · 2025-12-03
> **Response to Reviewer 2U6W, Part 4**
>
> $\textbf{Q6}$: Figure 2: What are the axes for the distribution plots? How many reservoir configurations were tested in total?
>
> $\textbf{A6}$: X-axis of the distribution plot represents the proportion of the model's correct predictions out of all its predictions, i.e. accuracy. 63 different combinations of reservoir hyperparameters were tested in total.
>
> &nbsp;
>
> $\textbf{Q7}$: Why use parametric UMAP instead of simpler alternatives like PCA or kernel PCA? What is lost if you use linear dimensionality reduction?
>
> $\textbf{A7}$: We use parametric UMAP because it provides a nonlinear dimensionality-reduction mapping that preserves the local geometry of high-dimensional feature spaces, as demonstrated in the original formulation of the method [McInnes, Leland, John Healy, and James Melville. "Umap: Uniform manifold approximation and projection for dimension reduction." arXiv preprint arXiv:1802.03426 (2018)]. The parametric version additionally allows learning an explicit neural-network mapping, which ensures stable and consistent embedding of new data during inference.
>
> We compared UMAP with simpler alternatives, including PCA and kernel PCA:
>
> | Metric                 | Accuracy                         | Recall                         | Precision                         | F1-Score                         |
> |---------------------------|-------------------------------|-------------------------------|-------------------------------|-------------------------------|
> | CSP-RC-CSP-UMAP-LDA | $\mathbf{0.87 \pm 0.05}$ | $\mathbf{0.80 \pm 0.10}$ | $\mathbf{0.88 \pm 0.09}$ | $\mathbf{0.83 \pm 0.07}$
> |CSP-RC-CSP-KPCA-LDA | $0.86 \pm 0.05$ | $0.72 \pm 0.12$ | $0.92 \pm 0.09$ | $0.80 \pm 0.08$
> |CSP-RC-CSP-PCA-LDA | $0.86 \pm 0.05$ | $0.72 \pm 0.10$ | $0.92 \pm 0.08$ | $0.80 \pm 0.07$
>
> These linear methods produced lower, but broadly comparable, performance. This result is consistent with the findings of Nozari et al., who showed that macroscopic brain dynamics are largely linear at the resting-state scale, and that linear autoregressive models often provide the best description of such aggregated neural signals [Nozari, Erfan, et al. "Macroscopic resting-state brain dynamics are best described by linear models." Nature biomedical engineering 8.1 (2024): 68-84].
>
> However, in our pipeline, the situation fundamentally changes after reservoir computing: the reservoir introduces nonlinear temporal transformations, producing a high-dimensional feature space with nonlinear structure. In this transformed space, linear dimensionality reduction such as PCA removes important geometric relationships and reduces class separability. In contrast, UMAP better preserves local structure and neighborhood relations, which translates into more stable and more accurate classification performance.
>
> Thus, UMAP is not chosen for complexity, but because it aligns with the nonlinear nature of the reservoir-generated features, whereas linear methods become limiting after the reservoir’s nonlinear temporal encoding.
>
> &nbsp;
>
> $\textbf{Q8}$: How sensitive is the pipeline to hyperparameters? Varying Minp, Mout, reservoir size, etc.?
>
> $\textbf{A8}$: We systematically investigated the influence of key hyperparameters on model performance, including number of input and output CSP components ($M_{inp}, M_{out}$), leak rate ($\gamma$), spectral radius ($\phi$). Our analysis revealed that initial increases in both $M_{inp}$ and $M_{out}$ progressively enhanced accuracy, with optimal performance achieved at $M_{inp}=14$ and $M_{out}=20$. Beyond these values, further increments did not improve pipeline performance, indicating saturation of beneficial effects.
>
> To assess the role of reservoir capacity, we evaluated model accuracy as a function of the number of reservoir neurons ($N_h \in [182,504]$, step 28). Accuracy gradually increased with larger reservoir sizes up to an optimal value of $N_h = 308$, after which additional neurons yielded negligible improvement. Based on this observation, the reservoir size was fixed at $N_h = 308$ in subsequent experiments.
>
> For the reservoir dynamics, we identified that leak rate and spectral radius do not greatly affect the classification accuracy: mean accuracy varies in range [0.82, 0.85] for whole investigated ranges of parameters.
> Comprehensive results of these parameter optimizations are provided in the Appendix.

---

### Meta-Review · Area_Chair_2oPj · 2025-12-18

**Summary:**

This paper proposes a multi-stage pipeline for resting-state fMRI classification that combines supervised spatial filtering via Common Spatial Patterns (CSP), reservoir computing for temporal encoding, parametric UMAP for nonlinear dimensionality reduction, and LDA for classification. The method applies CSP both to raw fMRI time series and again to reservoir states, with the goal of preserving and enhancing discriminative spatial structure through nonlinear temporal dynamics. The approach is evaluated primarily on a small in-house MDD dataset, with additional results reported on the ABIDE NYU autism dataset, and includes an interpretability analysis based on SHAP backprojection to brain regions.

Overall, the reviews are consistent in recognizing the paper’s strengths, namely, its interpretability, clear presentation of a complete pipeline, and relevance to clinical neuroimaging, while agreeing that these strengths are insufficient for acceptance at ICLR. The submission would likely be more appropriate for a domain-focused or applied machine learning venue, where careful integration of known methods and empirical performance on specific datasets are central evaluation criteria. For ICLR, however, the combination of limited novelty, constrained experimental validation, and weak alignment with core ML contributions justifies the reject decision.

**Reviewer Concerns:**

While the problem domain is relevant and the paper presents a carefully engineered and interpretable pipeline, the reviewers converge on the assessment that the submission does not meet the bar for acceptance at ICLR. The primary concern is the lack of sufficient novelty from a machine learning perspective. The proposed method is largely a composition of well-established techniques, CSP, reservoir computing, UMAP, and LDA, without introducing new algorithms, theory, or conceptual insights that would be broadly impactful to the ML community. The central idea of applying CSP to reservoir states, while interesting, is viewed as an incremental engineering variation rather than a substantive methodological contribution. As such, the work reads more like an application-focused study tailored to neuroimaging than a paper advancing core ML understanding.

The experimental validation is also not strong enough to support the paper’s claims, particularly those suggesting state-of-the-art performance. The main results rely on a relatively small dataset of 163 subjects, raising concerns about overfitting and generalization. Although cross-validation is used and additional ablations were added in response to reviews, the evaluation remains limited in scope. The ABIDE results, while helpful, are secondary and rely on comparisons to numbers reported in other papers rather than a unified experimental protocol with strong baselines implemented and tuned in the same framework. Reviewers consistently note the absence of competitive modern baselines, such as temporal CNNs, RNN-based models, transformer-based approaches, or recent representation learning methods that are now standard points of comparison for fMRI time-series classification. This makes it difficult to assess whether the reported performance gains stem from genuine methodological advantages or from weaknesses in the chosen baselines.

Several reviewers also point out that the paper lacks a clear theoretical or conceptual justification for key design choices, including the double application of CSP and the necessity of parametric UMAP after supervised spatial filtering. While the authors provide post hoc explanations and empirical ablations, the method remains largely motivated by empirical effectiveness rather than principled reasoning. This further reinforces the perception that the contribution is an ad hoc pipeline rather than a coherent ML model with insights transferable beyond the specific application.

**Reviewer Scores:**

Based on the content of the reviews and the subsequent author responses, it is unlikely that any reviewer would have substantially increased their score had they participated fully in the discussion.

Reviewer 2U6W would almost certainly have maintained a reject score. Their concerns focus on fundamental issues—lack of novelty, weak baselines, questionable evaluation practices, and insufficient theoretical grounding—that were only partially addressed and largely remain misaligned with ICLR’s expectations. Even with added ablations and clarifications, their core judgment that the work is not ICLR-level would not plausibly change.

Reviewer WaDm would also be expected to keep a reject score. This reviewer emphasized the application-heavy nature of the paper, insufficient experimental breadth, unclear validation protocol, and lack of generalization evidence. Although additional datasets and ablations were added, the work still does not provide the kind of robust, multi-dataset, modern-baseline evaluation this reviewer was asking for, making an upward score change unlikely.

Reviewer o33G would likely remain at reject as well. While acknowledging that the end-to-end pipeline is sensible and reasonably executed, this reviewer explicitly judged the novelty as limited and the evaluation as too narrow. The added ABIDE results and responses help, but do not fundamentally change the assessment that the paper does not contribute new ML ideas or fair comparisons against strong alternatives.

Reviewer tTvF, who initially gave a borderline score, might be the only reviewer whose score could plausibly increase slightly with full discussion. However, even in this case, it is more likely that they would remain at a marginal or weak accept-at-best position rather than clearly advocate for acceptance. Their review already notes that the paper is below the ICLR bar in novelty and ML contribution, and the discussion does not introduce new elements that decisively address those concerns.

---

### Decision · Program_Chairs · 2026-01-26

Reject